# Ventral hippocampal projections to the medial prefrontal cortex regulate social memory

**Mary L Phillips, Holly Anne Robinson, Lucas Pozzo-Miller\***

Department of Neurobiology, The University of Alabama at Birmingham, Birmingham, United States

**Abstract** Inputs from the ventral hippocampus (vHIP) to the medial prefrontal cortex (mPFC) are implicated in several neuropsychiatric disorders. Here, we show that the vHIP-mPFC projection is hyperactive in the *Mecp2* knockout mouse model of the autism spectrum disorder Rett syndrome, which has deficits in social memory. Long-term excitation of mPFC-projecting vHIP neurons in wild-type mice impaired social memory, whereas their long-term inhibition in Rett mice rescued social memory deficits. The extent of social memory improvement was negatively correlated with vHIP-evoked responses in mPFC slices, on a mouse-per-mouse basis. Acute manipulations of the vHIP-mPFC projection affected social memory in a region and behavior selective manner, suggesting that proper vHIP-mPFC signaling is necessary to recall social memories. In addition, we identified an altered pattern of vHIP innervation of mPFC neurons, and increased synaptic strength of vHIP inputs onto layer five pyramidal neurons as contributing factors of aberrant vHIP-mPFC signaling in Rett mice.

DOI: https://doi.org/10.7554/eLife.44182.001

## Introduction

Social interactions are a fundamental part of our daily lives, and impairments in social cognition are key features of multiple neuropsychiatric illnesses. A person or animal must reliably recall previous social interactions to make appropriate social responses and then update the memory with each new encounter. Previous studies have identified the hippocampal network as the brain region that tracks social encounters in both human subjects and mouse models (*Hitti and Siegelbaum, 2014*; *Meira et al., 2018*; *Okuyama et al., 2016*; *Tavares et al., 2015*). Functional neuroimaging in human subjects has revealed that higher covariance between hippocampal activity and changes in social environment reflect better social skills (*Tavares et al., 2015*). In mouse models, perturbing neuronal activity in both dorsal CA2 and ventral CA1 hippocampal subregions impairs social memory (*Hitti and Siegelbaum, 2014*; *Meira et al., 2018*; *Okuyama et al., 2016*). However, debate remains as to which long-range efferent projections from the hippocampus are required for the formation of social memories.

Autism spectrum disorders (ASDs) are characterized by difficulties in interpreting social situations and a lack of social appropriation (*Barendse et al., 2018*). A common feature in mouse models of monogenic ASDs is an imbalance in synaptic excitation and inhibition (E/I) within different brain regions (*Nelson and Valakh, 2015*). Altering the E/I balance in the medial prefrontal cortex (mPFC) of mice mimics autism-like social deficits (*Yizhar et al., 2011*), and restoring the E/I balance in the *CNTNAP2* knockout (KO) and valproate mouse models of ASDs improves their social deficits (*Brumback et al., 2018*; *Selimbeyoglu et al., 2017*). Excitatory pyramidal neurons of the ventral hippocampus (vHIP) send long-range projections to the mPFC (*Dégenètais et al., 2003*; *Dembrow et al., 2010*; *Liu and Carter, 2018*; *Thierry et al., 2000*), and the activity of different

\*For correspondence:
lucaspm@uab.edu

**Competing interests:** The authors declare that no competing interests exist.

populations of mPFC pyramidal neurons are correlated with the novelty of a social target (*Liang et al., 2018*). Thus, the mPFC is a prime candidate region for the relay of social memory-related signaling from the vHIP. Therefore, we tested whether altering the activity of mPFC-projecting vHIP neurons affects social behavior and memory, and if this long-range projection is dysfunctional in a mouse model of the monogenic syndromic ASD Rett syndrome (RTT). We focused on the *Mecp2* KO mouse model of RTT because of the heightened activity in the vHIP (*Calfa et al., 2011*; *Calfa et al., 2015*) and the hypoactivity of cortical regions (*Durand et al., 2012*; *Kron et al., 2012*; *Morello et al., 2018*; *Sceniak et al., 2016*; *Tomassy et al., 2014*), both resulting from opposite changes in their microcircuit E/I balance.

Using a combination of unbiased behavioral analyses, pathway-specific chemogenetic manipulations with an intersectional genetic approach, high-speed imaging of network activity with voltage-sensitive dyes, trans-synaptic tracing, intracellular recordings, and dual-color optogenetics, we showed that the long-range vHIP-mPFC projection is hyperactive in *Mecp2* KO mice, which results in social memory deficits. Furthermore, chemogenetic manipulation of mPFC-projecting vHIP neurons in wild-type (WT) and *Mecp2* KO mice correlated with social memory performance in a specific and selective manner. Lastly, these behavioral consequences arose from alterations in the morphology and function of excitatory synapses between vHIP axons and pyramidal neurons in layer 5 of the mPFC.

## Results

### mPFC-projecting vHIP neurons are selectively activated during social encounters

Because both the vHIP and mPFC have been independently implicated in different aspects of social behavior (*Hitti and Siegelbaum, 2014*; *Liang et al., 2018*; *Meira et al., 2018*; *Okuyama et al., 2016*; *Selimbeyoglu et al., 2017*; *Yizhar et al., 2011*), we first tested if the vHIP projection to the mPFC is selectively engaged during social encounters. To identify vHIP neurons based on their long-range projections, we bilaterally injected green RetroBeads into the prelimbic (PL) region of the mPFC and red RetroBeads into the lateral hypothalamus (LH) of male WT and *Mecp2* KO mice at postnatal day 31 (P31), and then we allowed 14 days for RetroBead transfer until P45, when male *Mecp2* KO mice are symptomatic (*Chen et al., 2001*) (*Figure 1A and B*). Test mice were placed in an open chamber and sequentially exposed for 10 min to either two inanimate objects (novel toy mice) or other live mice (a cage-mate WT littermate and an age-matched non-cage-mate WT mouse), with an interval of 1 hr between exposures (*Figure 1—figure supplement 1A*). Forty-five minutes after the last interaction, we perfused the mice and prepared their brains for quantitative immunohistochemistry of the immediate early gene c-Fos as an estimate of neuronal activity (*Cohen and Greenberg, 2008*), measuring the c-Fos intensity of each RetroBead-containing neuron (*Figure 1C–D*). All vHIP neurons in WT mice that interacted with live mice showed higher c-Fos intensities than those in mice that interacted with toy mice, irrespective of their efferent projections [p<0.0001, Three-Way ANOVA followed by Benjamini and Hochberg Multiple Comparisons (B and H-MC); *Figure 1D–E*]. Furthermore, mPFC-projecting vHIP neurons in WT mice that interacted with live mice had higher c-Fos intensities than those projecting to the LH (n = 275 mPFC-projecting neurons in six sections from four mice, n = 282/3/2 LH-projecting neurons; p=0.0022, Three-Way ANOVA followed by B and H-MC).

We obtained similar results in *Mecp2* KO mice, with higher c-Fos intensities in mPFC-projecting vHIP neurons of mice interacting with live mice compared to those interacting with toy mice (p=0.0034, Three-Way ANOVA followed by B and H-MC; *Figure 1D–E*). However, the difference between mPFC- and LH-projecting vHIP neurons from mice that interacted with live mice was not statistically significant, suggesting the loss of selectivity in vHIP signaling in *Mecp2* KO mice (p=0.1544, Three-Way ANOVA followed by B and H-MC; *Figure 1D–E*). In addition, the differences in c-Fos intensities between the toy mice and live mice conditions were smaller in *Mecp2* KO mice compared to WT mice, likely due to the higher basal activity in the vHIP of *Mecp2* KO mice (*Calfa et al., 2011*; *Calfa et al., 2015*). Consistent with the lower levels of neuronal activity reported previously in the mPFC (*Kron et al., 2012*), the PL region of the mPFC of *Mecp2* KO mice had fewer c-Fos-positive neurons than that of WT mice that interacted with live mice (p=0.0234, Two-Way

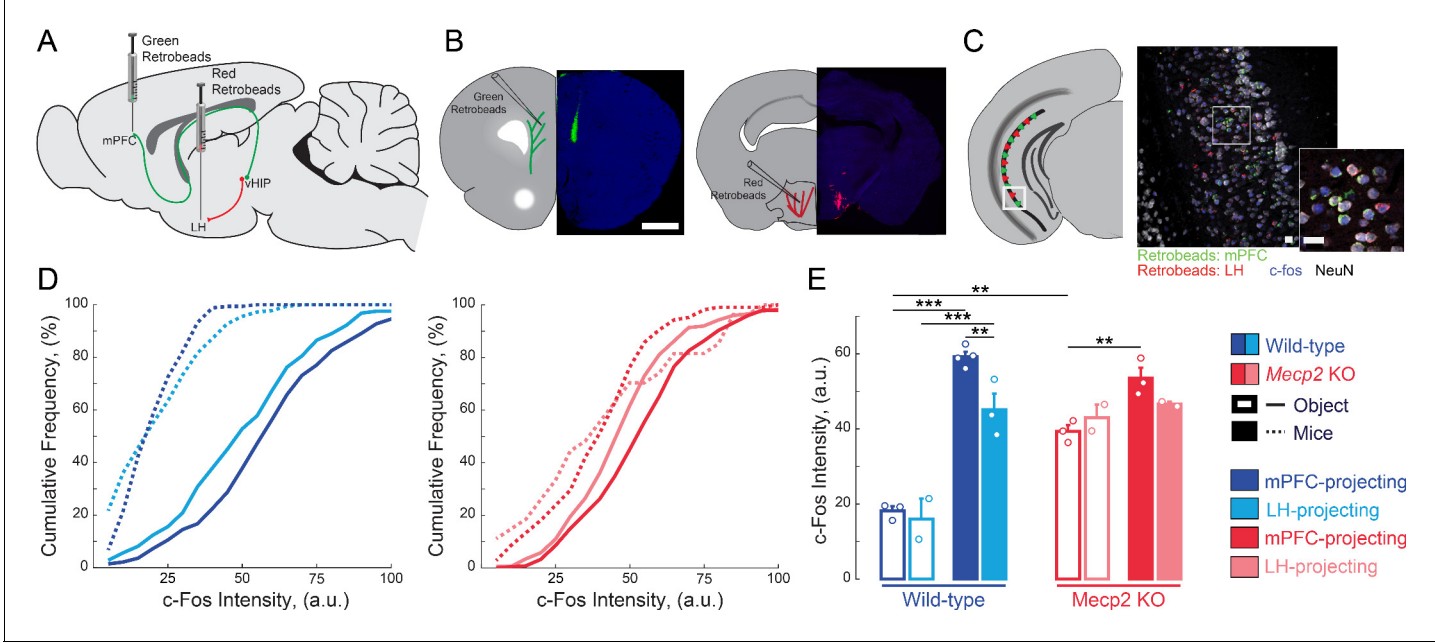

**Figure 1.** mPFC-projecting vHIP neurons are selectively activated during sequential social encounters with familiar and novel mice. (**A**) Schematic of RetroBead injection for labeling mPFC- or LH-projecting vHIP neurons. (**B**) Injection sites. Scale bar 1 mm. (**C**) RetroBead labeling and c-Fos immunohistochemistry in vHIP sections. Scale bar 25 μm. (**D**) Cumulative probability distributions of c-Fos intensities of RetroBead-labeled neurons [n = 163 cells from 3 sections from three mice (163/3/3) WT mPFC-projecting with toy mice; n = 180/3/2 WT LH-projecting with toy mice; n = 275/6/4 WT mPFC-projecting with live mice; n = 271/4/3 WT LH-projecting with live mice; n = 105/6/3 *Mecp2* KO mPFC-projecting with toy mice; n = 22/3/2 *Mecp2* KO LH-projecting with toy mice; n = 247/4/3 *Mecp2* KO mPFC-projecting with live mice; n = 172/5/2 *Mecp2* KO LH-projecting with live mice]. (**E**) Mean c-Fos intensity of RetroBead-labeled neurons, averaged per mouse. [WT mPFC toy mice vs. WT LH toy mice, p 0.6463; WT mPFC toy mice vs. WT mPFC live mice, p<0.0001; WT LH toy mice vs. WT LH live mice, p<0.0001; WT mPFC live mice vs. WT LH live mice, p=0.0022; KO mPFC toy mice vs. KO LH toy mice, p=0.4612; KO mPFC toy mice vs. KO mPFC live mice, p=0.0034; KO LH toy mice vs. KO LH live mice, p=0.4885; KO mPFC live mice vs. KO LH live mice, p=0.1544; WT mPFC toy mice vs. KO mPFC toy mice, p=0.0001; WT mPFC live mice vs. KO mPFC live mice, p=0.1544; Projection p=0.0292; Genotype p<0.0001; Experience p<0.0001; Projection x Genotype p=0.1263; Genotype x Experience p<0.0001; Projection x Genotype x Experience p=0.8545; Three Way ANOVA with Benjamini and Hochberg Multiple Comparisons]. Mean ± SEM; *p<0.05, **p<0.01. *Figure 1—source data 1*. See also *Figure 1—figure supplement 1*.

DOI: https://doi.org/10.7554/eLife.44182.002

The following source data and figure supplements are available for figure 1:

**Source data 1.** mPFC-projecting vHIP neurons are selectively activated during sequential social encounters with familiar and novel mice.
DOI: https://doi.org/10.7554/eLife.44182.005
**Figure supplement 1.** Higher number of c-Fos-positive neurons in the mPFC following social encounters.
DOI: https://doi.org/10.7554/eLife.44182.003
**Figure supplement 1—source data 1.** Higher number of c-Fos-positive neurons in the mPFC following social encounters.
DOI: https://doi.org/10.7554/eLife.44182.004

ANOVA followed by B and H-MC; *Figure 1—figure supplement 1B*). However, both WT and *Mecp2* KO mice that interacted with live mice had more c-Fos-positive neurons compared to those that interacted with toy mice, indicating that the mPFC of both genotypes is more robustly activated during a social encounter than by exposure to novel inanimate objects (p<0.05, Two-Way ANOVA followed by B and H-MC; *Figure 1—figure supplement 1*).

## Atypical social behavior and impaired social memory in *Mecp2* KO mice

To assess social behaviors, we used a three-chamber interaction arena to sequentially test social preference and then social memory. For social preference, we allowed mice to explore either a chamber containing a novel mouse (stranger 1) restrained under an inverted pencil cup or a chamber containing an empty inverted pencil cup (*Figure 2A*). Both WT and *Mecp2* KO mice spent more time investigating the cup containing stranger one compared to the empty cup. The discrimination index (DI) of this preference is statistically different than chance in both genotypes, and comparable

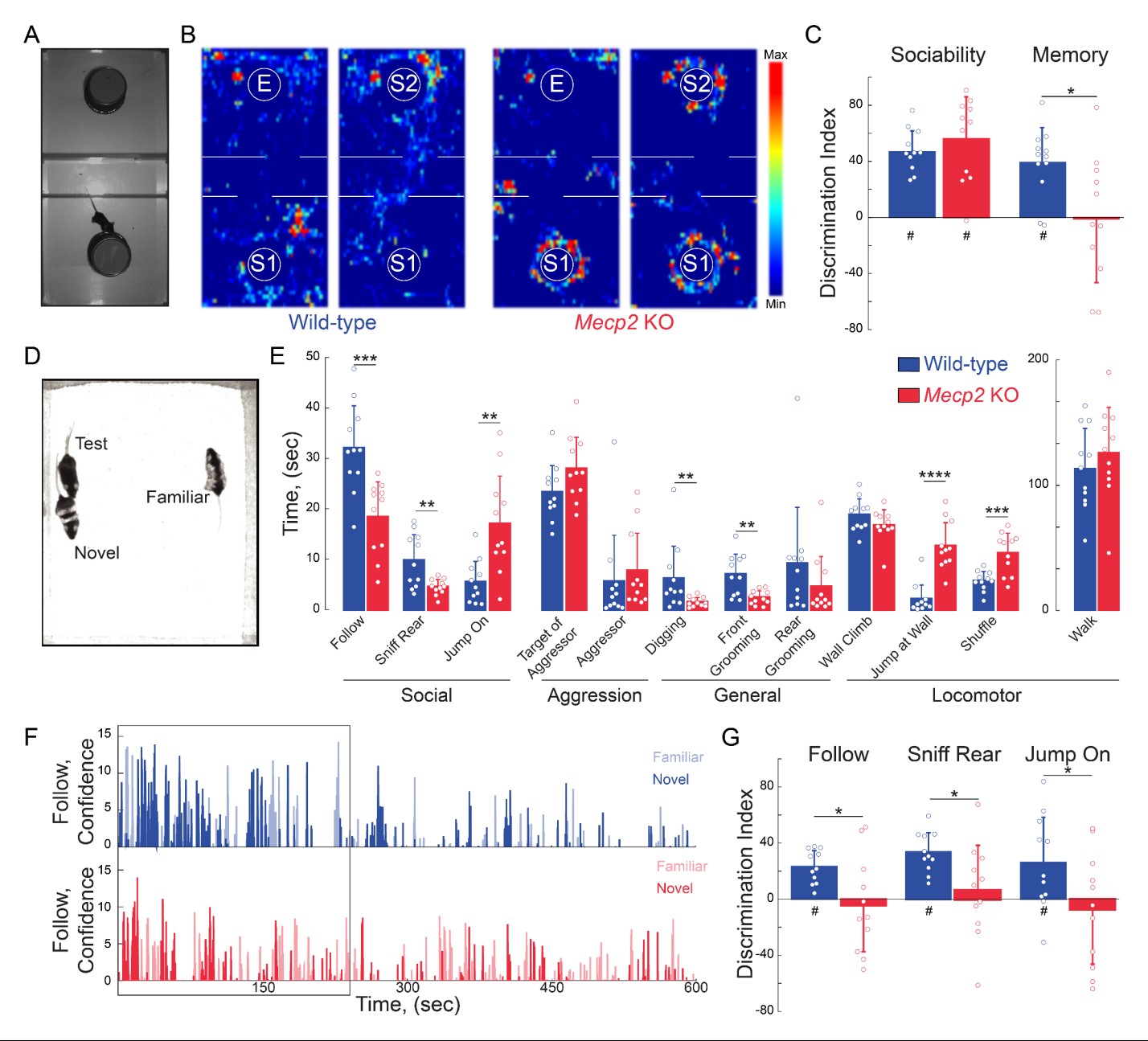

**Figure 2.** *Mecp2* KO mice have impaired social memory and atypical social interactions. (**A**) Schematic of three-chamber social test. (**B**) Representative heat maps of WT and *Mecp2* KO mice during the three-chamber social test. (**C**) Discrimination indices of sociability and memory tests. (n = 12 WT mice; n = 11 *Mecp2* KO mice; WT vs. *Mecp2* KO sociability, p=0.3523; WT vs. *Mecp2* KO, memory p=0.0140; Two-sample Student's t-test; WT sociability DI vs. chance, p<0.0001; *Mecp2* KO sociability DI vs. chance, p<0.0001; WT memory DI vs. chance, p=0.0002; *Mecp2* KO memory vs. chance, p=0.9436; One-sample Student's t-test). (**D**) Image of unrestricted social memory paradigm. (**E**) Time spent performing different behaviors using automated scoring of unrestricted social encounters (n = 11 WT mice; n = 11 *Mecp2* KO mice; Follow, p=0.0008; Rear Sniff, p=0.0055; Jump On, p=0.0020; Target of Aggressor, p=0.0857; Aggressor, p=0.1513; Digging, p=0.0052; Grooming Front, p=0.0025; Grooming Rear, p=0.1875; Wall Climb, p=0.1713; Wall Jump, p<0.0001; Shuffle, p=0.0007; Walk, p=0.4135; Student's t-test or Mann-Whitney test dependent on normalcy). (**F**) Computer confidence scores in following behavior over the course of representative videos separated by target: novel mouse in bold and familiar opaque, first 4 min boxed. (**G**) Time the test mouse spent following, sniffing rear, or 'jumping on' either the novel or familiar mouse during the first 4 min of the video (n = 11 WT; n = 11 KO mice; WT Follow DI vs. chance, p<0.0001; WT Rear Sniff, p<0.0001; WT Jump On, p=0.0279; *Mecp2* KO Follow p=0.6683, *Mecp2* KO Rear Sniff, p=0.5656; *Mecp2* KO Jump On, p=0.5665; Two-sample Student's t-test;WT vs. *Mecp2* KO Follow, p=0.0200; WT vs. *Mecp2* KO Sniff, p=0.0229; WT vs. *Mecp2* KO Jump On, p=0.0493; One-sample Student's t-test). Mean ± SD; *p<0.05, **p<0.01. For memory tasks (C and G), * indicate comparisons between genotypes while # indicates differences of the discrimination indices from chance value (0). *Figure 2—source data 1*. See also *Figure 2—figure supplements 1–2*.

*Figure 2 continued on next page*

*Figure 2 continued*

DOI: https://doi.org/10.7554/eLife.44182.006

The following source data and figure supplements are available for figure 2:

**Source data 1.** *Mecp2*KO mice have impaired social memory and atypical social interactions.

DOI: https://doi.org/10.7554/eLife.44182.010

**Figure supplement 1.** Impaired social memory in *Mecp2* KO mice.

DOI: https://doi.org/10.7554/eLife.44182.007

**Figure supplement 1—source data 1.** Impaired social memory in *Mecp2*KO mice.

DOI: https://doi.org/10.7554/eLife.44182.008

**Figure supplement 2.** WT mice show preference for novel mice during the first 4 min of unrestricted interactions.

DOI: https://doi.org/10.7554/eLife.44182.009

between the two genotypes (n = 12 WT mice; n = 11 *Mecp2* KO mice; One-sample t-test against chance p<0.0001 for both genotypes; Two-sample t-test p=0.3523; *Figure 2B–C* and *Figure 2—figure supplement 1A*). These data indicate that social preference is intact in *Mecp2* KO mice. Immediately following the sociability test (within 2 min), we placed a second novel mouse (stranger 2) under the previously empty pencil cup, and again allowed the test mice to explore both chambers. Indicative of social memory for stranger one and their preference for novel mice, WT mice spent more time investigating the cup containing stranger 2 (n = 12 WT mice, p=0.0002, One-sample t-test; *Figure 2B–C* and *Figure 2—figure supplement 1A*). However, the DI of *Mecp2* KO mice was significantly different than that of WT mice, and not statistically different than chance (n = 11 *Mecp2* KO mice; Two-sample t-test p=0.0140; One-sample t-test p=0.9436; *Figure 2B–C* and *Figure 2—figure supplement 1A*), indicating a deficit in the social memory of the stranger one mouse that was encountered 2 min before.

To avoid potential confounds arising from testing social interactions with mice restrained under pencil cups, we implemented a behavioral assay in which WT or *Mecp2* KO mice freely interacted simultaneously for 10 min with both a co-housed WT littermate and a novel age-matched WT mouse (*Figure 2D*). Unbiased scoring of social interactions by the machine-learning based Janelia Automatic Animal Behavior Annotator (*JAABA*) (*Kabra et al., 2013*; *Ohayon et al., 2013*; *Robie et al., 2017*) revealed that WT mice mainly engaged in following behavior. By contrast, *Mecp2* KO mice followed and sniffed other mice significantly less than WT mice, but engaged in an atypical 'jumping on' behavior (*Figure 2E*). In addition, *Mecp2* KO mice displayed less digging and facial grooming, more wall jumping, and more shuffled walking (n = 11 WT mice, n = 11 *Mecp2* KO mice, p<0.05, Student's t-test or Mann-Whitney test dependent on normalcy; *Figure 2E*). As a critical control for social interactions in the unrestricted arena and the three-chamber test, WT and *Mecp2* KO mice spent a comparable amount of time walking (p=0.4135, Student's t-test; *Figure 2E*). Overall, these data indicate that, while *Mecp2* KO mice do display interest in other mice, they do so in an atypical manner.

During the first 4 min of the unrestricted assay, WT mice preferentially interacted with the novel mouse across all measured social behaviors, including following, sniffing, and 'jumping on' (n = 11 mice, p<0.05, One-sample t-test; *Figure 2G* and *Figure 2—figure supplement 1B*), after which social interaction declined regardless of the target mouse (*Figure 2F* and *Figure 2—figure supplement 2*). The discrimination indices between WT mice and *Mecp2* KO mice were significantly different (p<0.05, Two-sample t-test; *Figure 2G*), and indeed *Mecp2* KO mice showed no significant preference between the familiar and novel mouse (n = 11 mice, p>0.05, One-sample; *Figure 2G* and *Figure 2—figure supplement 1B*), consistent with previously observed deficits in social memory (see *Figure 2C*).

## Increased influence of vHIP inputs on the mPFC network in *Mecp2* KO mice

Because mPFC-projecting vHIP neurons are activated during social encounters and *Mecp2* KO mice have impaired social memory and atypical social behaviors, we next characterized vHIP inputs to the mPFC in *Mecp2* KO mice at the functional level. To identify hippocampal fibers in ex vivo slices of the mPFC, we injected the fluorescent tracer dextran-Alexa-594 into the vHIP and, after 2 weeks to

allow its anterograde transfer, prepared brain slices at a 10° angle from the coronal plane (*Parent et al., 2010*) (*Figure 3A*). We evoked field excitatory postsynaptic potentials (fEPSPs) with single monopolar current pulses (100 µs) delivered through a theta-glass electrode placed in the fluorescently labeled vHIP fiber bundle and simultaneously imaged voltage-sensitive dye (VSD) signals, which are directly proportional to the slope of individual fEPSPs and follow their rise and decay kinetics (*Grinvald et al., 1988*; *Chang and Jackson, 2003*; *Bandyopadhyay et al., 2005*; *Calfa et al., 2011*) (*Figure 3B* and *Figure 3—figure supplement 1*). The amplitudes of VSD signals evoked by single pulse stimulation of vHIP afferents were larger in mPFC slices from *Mecp2* KO mice compared to WT littermates at a range of stimulation intensities [n = 11 slices from 7 WT mice, n = 11/5 *Mecp2* KO mice, p=0.0470, Two-way repeated measures (RM) ANOVA; *Figure 3D and E*]. The spatiotemporal spread of VSD signals throughout the mPFC slice was similar between *Mecp2* KO and WT mice (p=0.4529, Two-way RM ANOVA; *Figure 3F*), although the spatial spread was significantly larger at lower stimulation intensities in slices from *Mecp2* KO mice (p=0.0133, Mann-Whitney test; *Figure 3G*). Stimulating intracortical fibers with another theta glass electrode placed in layer 2/3 of the same cortical column evoked VSD amplitudes of comparable amplitude in slices from *Mecp2* KO and WT mice (n = 11 slices from 7 WT mice, n = 11/5 *Mecp2* KO mice, p=0.0540; Two-way RM ANOVA; *Figure 3H and I*). By contrast, the spatiotemporal spread of VSD signals evoked by intracortical stimulation was significantly smaller in slices from *Mecp2* KO mice (p=0.0498, Two-way RM ANOVA; *Figure 3J and K*). The amplitude and spatial spread of VSD signals evoked by vHIP stimulation were 72% and 71%, respectively, of those evoked by intracortical stimulation in mPFC slices from WT mice. However, the amplitude and spatial spread of VSD signals evoked by vHIP stimulation were 95% and 96% of those evoked by intracortical stimulation in *Mecp2* KO slices, which reflects both larger vHIP-evoked signals and smaller responses to intracortical stimulation (*Figure 3L–3N*). These data indicate that vHIP fibers drive hyperactivation of the mPFC network in *Mecp2* KO mice, in contrast to the hypoactivation driven by intracortical stimulation, suggesting that vHIP inputs are overrepresented in the mPFC network of *Mecp2* KO mice.

Considering the role of long-term synaptic plasticity in memory, we tested the ability of excitatory vHIP-mPFC synapses to undergo long-term potentiation (LTP) in slices from *Mecp2* KO mice, because previous studies have described plasticity at these synapses in rats and mice in vivo (*Izaki et al., 2003*; *Izaki et al., 2001*; *Laroche et al., 2000*). In mPFC slices from WT mice, high-frequency stimulation of vHIP afferents evoked a significant potentiation of the spatiotemporal spread of VSD signals, which persisted up to 45 min (n = 10 slices from seven mice, p=0.0013, Student's paired t-test, *Figure 3—figure supplement 1D*) and was sensitive to the N-methyl-D-aspartate receptor (NMDAR) antagonist APV (100 µM) (n = 4 slices from four mice, p=0.9205; Student's paired t-test; *Figure 3—figure supplement 1F*). However, mPFC slices from *Mecp2* KO mice showed only a short-term enhancement of the spatiotemporal spread of VSD signals, which quickly decayed back to baseline levels (n = 9 slices from five mice, p=0.2705; Student's paired t-test; *Figure 3—figure supplement 1E*). These data demonstrate an impairment of LTP at excitatory vHIP-mPFC synapses, similar to that previously reported at CA3-CA1 synapses in hippocampal slices of *Mecp2* KO mice (*Li et al., 2016*).

## Selective chemogenetic manipulation of mPFC-projecting vHIP neurons regulates social memory

To causally link the enhanced vHIP input to the mPFC with the deficits in social behavior in *Mecp2* KO mice, we used an intersectional genetic approach to express 'designer receptors exclusively activated by designer drugs' (DREADDs) selectively in mPFC-projecting vHIP neurons, and then modulate their activity with the designer ligand clozapine-N-oxide (CNO) (*Armbruster et al., 2007*; *Boender et al., 2014*). We injected a retrogradely transported canine adenovirus-2 (CAV-2) expressing Cre recombinase (Cre; CAV-2-Cre) bilaterally into the mPFC. We then injected adeno-associated virus serotype 8 (AAV8) expressing either excitatory (hM3Dq) or inhibitory (hM4Di) DREADDs from a Cre-dependent double-floxed inverse open reading frame (DIO) (*Hnasko et al., 2006*; *Kremer et al., 2000*) bilaterally in the vHIP of WT and *Mecp2* KO mice at P20 (*Figure 4A*). Control mice injected with CAV2-Cre and AAV8-DIO-mCherry were also treated with CNO to account for potential peripheral conversion of CNO into clozapine (*Gomez et al., 2017*). This intersectional approach resulted in sparse labeling of pyramidal neurons in the ventral CA1 region with their axons projecting to the mPFC (*Figure 4B*). To modulate the vHIP-mPFC circuit in a long-term manner, we

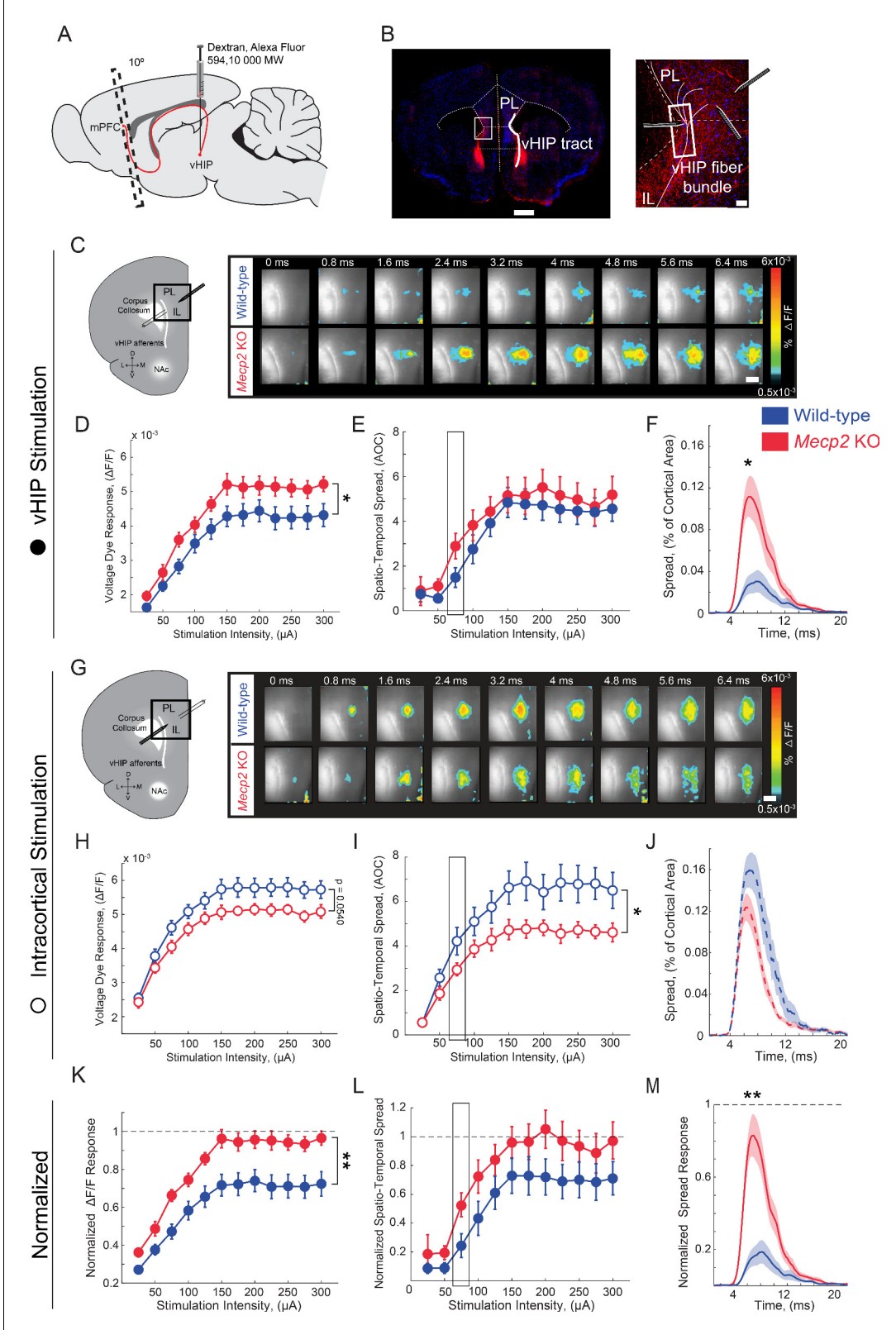

**Figure 3.** Increased influence of vHIP axons in the mPFC of *Mecp2* KO mice. (**A**) Schematic of dextran injection into the vHIP. (**B**) Visualization of vHIP fibers in mPFC slices. Scale bar 0.5 mm; inset 100 µm. (**C**) VSD responses are proportional to the amplitude and follow the kinetics of fEPSPs. (**D**) Representative VSD responses evoked by stimulation of fluorescently-labeled vHIP fibers. Scale bar 100 µm (D and H). (**E–G**) Input-output relationship of peak VSD responses (**E**) p=0.047, Two-way ANOVA), spatiotemporal spread (**F**) p=0.4529, Two-way ANOVA), and spread over time (**G**) p=0.0133,

*Figure 3 continued on next page*

*Figure 3 continued*

Mann-Whitney) (at 75 µA intensity) evoked by vHIP fiber stimulation. (H) Representative VSD responses evoked by intracortical stimulation. (I–K) Input-output relationship of peak VSD responses (I) p=0.4553, Two-way ANOVA), spatiotemporal spread (J) p=0.0498, Two-way ANOVA), and spread over time (K) p=0.1025, Student's t-test) (at 75 µA intensity) evoked by intracortical stimulation. (L–N) Peak VSD responses (L) p=0.0015, Two-way ANOVA), spatiotemporal spread (M) p=0.0767, Two-way ANOVA), and spread over time (N) p=0.0002, Mann-Whitney) evoked by vHIP fiber stimulation normalized to those evoked by intracortical stimulation. [n = 11 slices from seven mice (11/7) WT mice; n = 11/5 *Mecp2* KO]. Spatiotemporal spread = AOC created by spread of the cortical area (% of total) and time (ms). Mean ± SEM; *p<0.05, **p<0.01. *Figure 3—source data 1*. See also *Figure 3—figure supplements 1–2*.

DOI: https://doi.org/10.7554/eLife.44182.011

The following source data and figure supplements are available for figure 3:

**Source data 1.** Increased influence of vHIP axons in the mPFC of *Mecp2*KO mice.

DOI: https://doi.org/10.7554/eLife.44182.015

**Figure supplement 1.** Input-output curves of VSD signal amplitudes and initial slope of fEPSP.

DOI: https://doi.org/10.7554/eLife.44182.012

**Figure supplement 2.** Impaired LTP at vHIP-mPFC synapses in *Mecp2* KO mice.

DOI: https://doi.org/10.7554/eLife.44182.013

**Figure supplement 2—source data 1.** Impaired LTP at vHIP-mPFC synapses in *Mecp2*KO mice.

DOI: https://doi.org/10.7554/eLife.44182.014

delivered CNO via the drinking water (5 mg/kg/day) (*Carvalho Poyraz et al., 2016*) beginning at P34 and continuing until we used mice for experiments (*Figure 4A*). Such long-term activation of DREADD receptors via CNO has been validated previously, with neurons expressing the excitatory DREADD hM3Dq in slices from mice exposed to CNO for 14 days still showing an increased firing rate after CNO application (*Cheng et al., 2019*). At the age of viral injections and the start of CNO treatment, *Mecp2* KO mice lack the behavioral and cellular features that will develop into Rett-like symptoms after P45 (*Calfa et al., 2011*; *Durand et al., 2012*; *Tomassy et al., 2014*). In addition, we confirmed that P20-25 *Mecp2* KO mice performed at WT levels in terms of social memory (*Figure 4—figure supplement 1*), VSD signals in mPFC evoked by either vHIP or intracortical stimulation (*Figure 4—figure supplement 2A–K*), and LTP at vHIP-mPFC synapses (*Figure 4—figure supplement 2L–O*). Notably, the amplitude of vHIP-evoked VSD signals in mPFC slices from *Mecp2* KO mice did not show the typical developmental reduction between P20-25 and P45-50 observed in WT slices, resulting in significantly larger responses in symptomatic *Mecp2* KO mice compared to age-matched WT mice (*Figure 4—figure supplement 3*), similar to CA3-evoked VSD responses in CA1 of hippocampal slices (*Calfa et al., 2011*; *Li et al., 2016*).

When tested in the unrestricted social assay, P45 WT mice expressing only the marker mCherry in mPFC-projecting vHIP neurons and treated with CNO for 11 days had a significant preference for the novel mouse (n = 10, p=0.00.0202; One-sample t-test against chance; *Figure 4C*), similar to naive untreated WT mice. CNO-treated WT mice expressing the excitatory DREADD hM3Dq in mPFC-projecting vHIP neurons (to mimic vHIP hyperactivity in *Mecp2* KO mice) had a significantly lower DI, which was not different than chance (n = 12, One-way ANOVA followed by B and H-MC, p=0.0287; One-sample t-test, p=0.6905; *Figure 4C*), indicating a deficit in social memory resembling that of *Mecp2* KO mice. Chronic inhibition of mPFC-projecting vHIP neurons with the inhibitory DREADD hM4Di also caused a significant decrease in the DI and impaired social memory in WT mice (n = 9, One-way ANOVA followed by B and H-MC, p=0.0081; One-sample t-test, p=0.3332;) (*Figure 4C*), underscoring the role of this long-range projection in social memory.

To further define the consequences of altered vHIP-mPFC signaling in social behaviors, we inhibited mPFC-projecting vHIP neurons in *Mecp2* KO mice with the inhibitory DREADD hM4Di and CNO administration from P34 until P45 to selectively reduce the hyperactivation of this long-range circuit. This manipulation was sufficient to increase the time following the novel mouse compared to the familiar littermate in 73% of the treated *Mecp2* KO mice, resulting in a significant preference for the novel mouse, and indicating a rescue of social memory (n = 15, One-sample t-test, p=0.0312, *Figure 4C* and *Figure 4—figure supplement 4A*). This DI was significantly higher than control CNO-treated *Mecp2* KO mice expressing mCherry, which did not discriminate between the novel mouse and the familiar littermate (n = 10, One-way ANOVA followed by B and H-MC, p=0.0287; One-sample t-test, p=0.1322; *Figure 4C* and *Figure 4—figure supplement 4A*), similar to naive

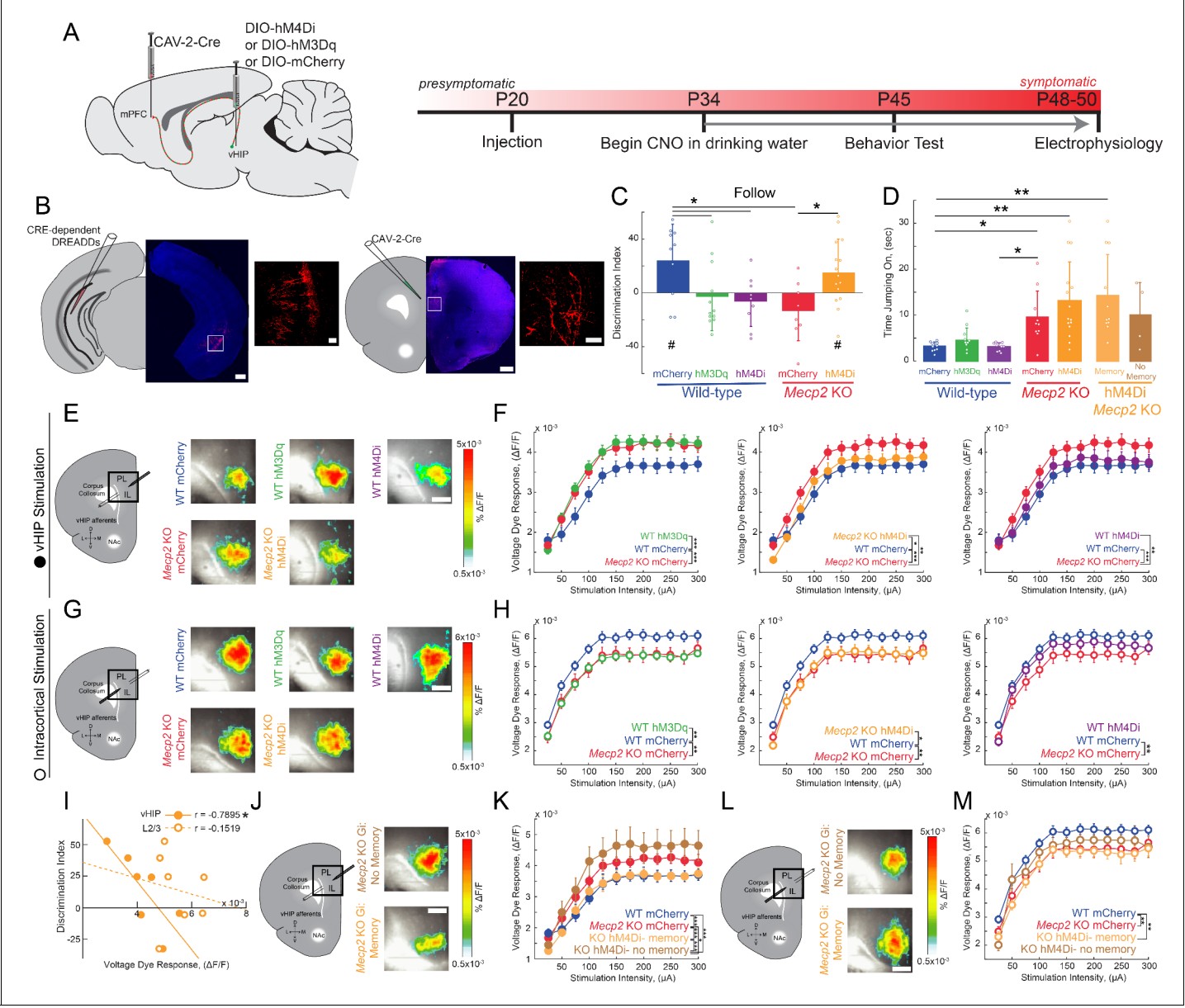

**Figure 4.** Activity of mPFC-projecting vHIP neurons modulates social memory in WT and *Mecp2* KO mice. (**A**) Schematic of CAV2-Cre and DREADD injections and experimental timeline. (**B**) Injection sites show sparse mCherry labeling of vHIP neurons with identifiable axons in the mPFC. Scale bar 500 μm large, 100 μm inset. (**C**) Discrimination Index of following familiar versus novel mice in unrestricted social interaction, scored by *JAABA* (n = 10 mCherry WT mice, p=0.0202; n = 12 hM3Dq WT mice, p=0.6905; n = 9 hM4Di WT mice, p=0.3332; n = 10 mCherry *Mecp2* KO, p=0.1322; n = 15 *Mecp2* KO mice, p=0.0312; One-sample Student's t-test against chance. mCherry WT vs. hM3Dq WT mice, p=0.0287; mCherry WT vs. hM4Di WT mice, p=0.0081; mCherry WT vs. mCherry *Mecp2* KO mice, p=0.0187; mCherry *Mecp2* KO vs. hM3Dq WT mice p=0.4674; mCherry *Mecp2* KO vs. hM4Di WT mice, p=0.6108; mCherry *Mecp2* KO mice vs. hM4Di *Mecp2* KO mice, p=0.0287; ANOVA p=0.0048; One-Way ANOVA with Benjamini and Hochberg Multiple Comparisons). (**D**) Time spent 'jumping on' other mice during unrestricted social interaction (n = 9 mCherry WT mice; n = 12 hM3Dq WT mice; n = 9 hM4Di WT mice; n = 10 mCherry *Mecp2* KO; n = 15 hM4Di *Mecp2* KO mice; mCherry WT vs. mCherry *Mecp2* KO, p=0.0215; hM3Dq WT vs. mCherry WT, p>0.9999; hM3Dq WT vs. hM4Di WT, p>0.9999; hM3Dq WT vs. mCherry *Mecp2* KO, p=0.2119; hM4Di WT vs. mCherry WT, p>0.9999; hM4Di WT vs. mCherry *Mecp2* KO, p=0.0194; mCherry *Mecp2* KO vs. hM4Di (All) *Mecp2* KO, p>0.9999; hM4Di (All) *Mecp2* KO vs. mCherry WT, p=0.0016; Memory hM4Di vs. mCherry KO, p>0.9999; Memory hM4Di vs. mCherry WT, p=0.0012; No Memory hM4Di vs. mCherry *Mecp2* KO, p>0.9999; No Memory hM4Di vs. mCherry WT, p=0.8023; Memory hM4Di vs. No Memory hM4Di, p>0.9999, Kruskal-Wallis test with Dunn's multiple corrections). (**E**) Representative VSD responses evoked by vHIP fiber stimulation in CNO-treated mice. Scale bar 200 μm. (**F**) Input-output relationships of peak VSD responses evoked by vHIP fiber stimulation. (mCherry WT vs. mCherry *Mecp2* KO, p<0.001; hM3Dq WT vs. mCherry WT, p<0.0001; hM3Dq WT vs. mCherry *Mecp2* KO, p=0.6530; mCherry WT vs. hM4Di, p=0.0642; hM4Di WT vs. mCherry *Mecp2* KO, p=0.0013; hM4Di *Mecp2* KO vs. mCherry WT, p=0.0333; hM4Di *Mecp2* KO vs. mCherry *Mecp2* KO, p=0.0016; Interaction p<0.0001; Stim p<0.0001; Group p=0.0193; Two-way RM

*Figure 4 continued on next page*

*Figure 4 continued*

ANOVA with Benjamini and Hochberg Multiple Comparisons). (G) Representative VSD responses evoked by intracortical stimulation in CNO-treated mice. (H) Input-output relationships of peak VSD responses evoked by intracortical stimulation (mCherry WT vs. mCherry *Mecp2* KO, p<0.0001; hM3Dq WT vs. mCherry WT, p<0.0001; hM3Dq WT vs. mCherry *Mecp2* KO, p 0.2164; mCherry WT vs. hM4Di WT, p=0.8296; hM4Di WT vs. mCherry *Mecp2* KO, p<0.0001; hM4Di *Mecp2* KO vs. mCherry WT, p=0.0023; hM4Di *Mecp2* KO vs. mCherry *Mecp2* KO, p=0.0726; Interaction p=0.2985; Stim p<0.0001; Group p=0.0479; Two-way RM ANOVA with Benjamini and Hochberg Multiple Comparisons). (I) Correlation between social memory DIs and VSD responses evoked by either vHIP fiber (closed circles) or intracortical stimulation (open circles) (n = 7 slices from 7 hM4Di *Mecp2* KO mice; Spearman r correlation; r = −0.7895, p=0.0347 vHIP fiber; r = −0.1519, p=0.7451 intracortical). (J–K) Input-output relationships of peak VSD responses evoked by vHIP fiber stimulation in slices from hM4Di *Mecp2* KO mice with intact or impaired social memory (Memory vs. mCherry WT, p=0.3043; Memory vs. mCherry *Mecp2* KO, p<0.0001; No Memory vs. mCherry WT, p<0.0001; No Memory vs. mCherry *Mecp2* KO, p=0.0406; Memory vs. No Memory, p<0.0001; Interaction p<0.0001; Stim p<0.0001; Group p=0.0056; Two-way RM ANOVA with Benjamini and Hochberg Multiple Comparisons). (L–M) Input-output relationships of peak VSD responses evoked by intracortical stimulation in slices from hM4Di *Mecp2* KO mice with intact or impaired social memory (Memory vs. mCherry WT, p=0.0002; Memory vs. mCherry *Mecp2* KO, p=0.3043; No Memory vs. mCherry WT, p=0.2171, No Memory vs. mCherry *Mecp2* KO, p=0.0720; Memory vs. No Memory, p=0.2171; Interaction p=0.2241; Stim p<0.0001; Group p=0.0761; Two-way RM ANOVA with Benjamini and Hochberg Multiple Comparisons). (E–M) n = 18 slices from 9 mCherry (18/9) WT mice; n = 12/8 hM3Dq WT mice; n = 16/9 hM4Di WT mice; n = 17/8 mCherry *Mecp2* KO mice; n = 20/10 hM4Di *Mecp2* KO mice; n = 11/6 hM4Di *Mecp2* KO memory mice; n = 6/4 hM4Di *Mecp2* KO no memory mice. (C–D) Mean ± SD; (F,H,K,M) Mean ± SEM; *p<0.05, **p<0.01. *Figure 4—source data 1*. See also *Figure 4—figure supplements 1–5*.
DOI: https://doi.org/10.7554/eLife.44182.016

The following source data and figure supplements are available for figure 4:

**Source data 1.** VSD responses to vHIP stimulation in mPFC slices and LTP at vHIP-mPFC synapses are not altered in presymptomatic *Mecp2*KO mice.
DOI: https://doi.org/10.7554/eLife.44182.027

**Figure supplement 1.** Social memory is not altered in presymptomatic *Mecp2* KO mice.
DOI: https://doi.org/10.7554/eLife.44182.017

**Figure supplement 1—source data 1.** VSD responses to vHIP stimulation in mPFC slices and LTP at vHIP-mPFC synapses are not altered in presymptomatic *Mecp2*KO mice.
DOI: https://doi.org/10.7554/eLife.44182.018

**Figure supplement 2.** VSD responses to vHIP stimulation in mPFC slices and LTP at vHIP-mPFC synapses are not altered in presymptomatic *Mecp2* KO mice.
DOI: https://doi.org/10.7554/eLife.44182.019

**Figure supplement 2—source data 1.** VSD responses to vHIP stimulation in mPFC slices of *Mecp2*KO mice do not develop in a typical manner.
DOI: https://doi.org/10.7554/eLife.44182.020

**Figure supplement 3.** VSD responses to vHIP stimulation in mPFC slices of *Mecp2* KO mice do not develop in a typical manner.
DOI: https://doi.org/10.7554/eLife.44182.021

**Figure supplement 3—source data 1.** Lack of effects of long-term DREADD stimulation by CNO on anxiety and general (non-social memory) behaviors.
DOI: https://doi.org/10.7554/eLife.44182.022

**Figure supplement 4.** Lack of effects of long-term DREADD stimulation by CNO on anxiety and general (non-social memory) behaviors.
DOI: https://doi.org/10.7554/eLife.44182.023

**Figure supplement 4—source data 1.** Memory Discrimination correlates to vHIP input to the mPFC in *Mecp2*KO mice.
DOI: https://doi.org/10.7554/eLife.44182.024

**Figure supplement 5.** Memory Discrimination correlates to vHIP input to the mPFC in *Mecp2* KO mice.
DOI: https://doi.org/10.7554/eLife.44182.025

**Figure supplement 5—source data 1.** Activity of mPFC-projecting vHIP neurons modulates social memory in WT and *Mecp2*KO mice.
DOI: https://doi.org/10.7554/eLife.44182.026

untreated *Mecp2* KO mice. In addition, there were no changes in the types of social behaviors displayed by CNO-treated mice expressing DREADDs in mPFC-projecting vHIP neurons. For example, the amount of time performing the atypical 'jumping on' behavior did not differ between hM4Di-expressing and mCherry control *Mecp2* KO mice (p>0.9999 within genotypes; p<0.05 between genotypes; Kruskal-Wallis test followed by Dunn's Multiple Comparisons; *Figure 4D*), suggesting that the vHIP-mPFC projection plays a specific role in the memory aspect of social interactions. Other than a small reduction of walking time in hM3Dq-expressing WT mice (p=0.0296, One-way ANOVA), there were no differences in grooming behavior, locomotion, or anxiety-like behaviors between DREADD-expressing WT and *Mecp2* KO mice and their mCherry-expressing controls after CNO treatment (p>0.05; *Figure 4—figure supplement 4B–F*).

After allowing 3–4 days for the potential effects of behavioral testing to fade, we prepared ex vivo mPFC slices from CNO-treated DREADD- and mCherry-expressing mice for VSD imaging. Consistent with their deficit in social memory, WT mice expressing the excitatory DREADD hM3Dq in mPFC-projecting vHIP neurons had larger vHIP-induced VSD signals in mPFC slices compared to mCherry-expressing WT controls, resembling those observed in mCherry-expressing *Mecp2* KO controls (n = 18 slices from 9 mCherry WT mice; n = 12/8 hM3Dq WT; n = 17/8 *Mecp2* KO mice; p<0.0001; p=0.6530; Two-way RM ANOVA; *Figure 4E–F*). Surprisingly, inhibiting mPFC-projecting vHIP neurons with hM4Di did not affect vHIP-induced VSD signals in mPFC slices of WT mice (p=0.0642; *Figure 4E-F*). While unexpected, we cannot rule out the possibility that other afferents may have increased their input to the mPFC, as reported previously (*Guirado et al., 2016*), or that other homeostatic mechanisms might have maintained proper activity levels in the mPFC. Interestingly, we still observed impaired memory performance, suggesting that dysfunction of the vHIP-mFPC pathway is sufficient to impair memory formation in the absence of large-scale changes to the mPFC network. As a control, mCherry expression in mPFC-projecting vHIP neurons followed by CNO treatment did not alter the difference in the amplitude of vHIP-induced VSD signals in mPFC slices between WT and *Mecp2* KO mice (p<0.0001; *Figure 4F*), which resemble those in naive untreated mice (see *Figure 3E*).

Because 27% of *Mecp2* KO mice expressing the inhibitory DREADD hM4Di in mPFC-projecting vHIP neurons did not show an improvement of social memory in the unrestricted test (see *Figure 4C*), we correlated their social memory DI with the amplitude of vHIP-induced VSD signals in mPFC slices on a mouse-per-mouse basis. This analysis uncovered a statistically significant negative correlation between DI and vHIP-induced mPFC responses (n = 7 pairs, r = −0.7895 p=0.0347, Spearman r correlation; *Figure 4I*). This correlation was also observed when data from mCherry- and hM4Di-expressing *Mecp2* KO mice were pooled together, indicating that the extent of dysfunction in the vHIP-mPFC projection underlies social memory impairments. Interestingly, this correlation is not statistically significant in any of the WT groups, which suggest an intriguing contribution of the altered mPFC microcircuit in *Mecp2* KO mice (*Figure 4—figure supplement 5*). Furthermore, vHIP-induced VSD signals in mPFC slices from *Mecp2* KO mice that showed improved social memory after expression of hM4Di in mPFC-projecting vHIP neurons were smaller than those in mCherry-expressing *Mecp2* KO controls (n = 11/6, p<0.0001, Two-way ANOVA; *Figure 4J–K*), resembling those in mCherry-expressing WT mice (p=0.3043). By contrast, vHIP-induced VSD signals in mPFC slices from *Mecp2* KO mice that still showed deficits in social memory after expression of hM4Di in mPFC-projecting vHIP neurons were significantly larger than those in mCherry-expressing WT mice (p<0.0001; Two-way ANOVA; *Figure 4J–K*). In addition, vHIP-evoked VSD signals in the mPFC of hM4Di-expressing *Mecp2* KO mice showing improved social memory DI were also larger than mCherry *Mecp2* KO controls (p=0.0406). Other than these effects on the social memory DI, no behavioral differences were observed between the hM4Di-expressing *Mecp2* KO mice with improved and impaired memory (*Figure 4C–D* and *Figure 4—figure supplement 4*).

Selective chemogenetic excitation of mPFC-projecting vHIP neurons in WT mice from P34 to P45 also affected VSD signals evoked by intracortical stimulation in layer 2/3 of mPFC slices. VSD responses in hM3Dq-expressing WT mice were significantly smaller than those in mCherry-expressing WT mice, and resemble those in mCherry-expressing *Mecp2* KO mice (n = 18 slices from 9 mCherry WT mice; n = 12/8 hM3Dq WT; n = 17/8 *Mecp2* KO mice; p=0.0004; p=0.8522; Two-way RM ANOVA, *Figure 4G–H*). However, chemogenetic inhibition with hM4Di did not affect VSD signals evoked by intracortical stimulation in WT mice (n = 16/9, p=0.2741, *Figure 4G–H*). VSD signals evoked by intracortical stimulation in mPFC slices from *Mecp2* KO mice expressing hM4Di in mPFC-projecting vHIP neurons were not significantly different than those in mCherry-expressing *Mecp2* KO controls, and were smaller than those in mCherry-expressing WT mice (n = 20/10; p=0.0311; p=0.5691; *Figure 4G–H*). Importantly, there was no significant correlation between the social memory DI and the amplitude of VSD signals evoked by intracortical stimulation in hM4Di-expressing *Mecp2* KO mice (n = 7 pairs, r = −0.1519, p=0.7451, Spearman r correlation; *Figure 4I*), and no difference in these signals between *Mecp2* KO mice that showed improvement in social memory and those that did not (n = 11/6 Memory; n = 6/4 No memory; p=0.3359; Two-way RM ANOVA; *Figure 4L–M*). Combined, these results indicate that selective chemogenetic modulation of mPFC-projecting vHIP neurons has specific consequences on the functional strength of this projection, but has smaller effects on other afferent inputs to the mPFC recruited by intracortical stimulation.

We next acutely manipulated neuronal activity to test whether the vHIP-mPFC projection is required for social memory recall, as opposed to the maintenance or initial formation of a social memory. We removed littermate sentinels from the home cage and administered a single intraperitoneal (i.p.) injection of CNO (1 mL/0.5 mg/100 g body weight) 2 hr before the unrestricted social test (*Figure 5A*). These i.p. CNO injections did not affect social memory in mCherry-expressing WT mice as their DI was significantly higher than chance (n = 10, p=0.009, One-sample t-test), whereas acute excitation of mPFC-projecting vHIP neurons by CNO activation of hM3Dq impaired social memory in WT mice (n = 10 hM3Dq WT mice, p=0.7775, One-sample t-test; *Figure 5B* and *Figure 5—figure supplement 1*). However, the DI was not statistically different between controls and hM3Dq-expressing WT mice in the mPFC projections group due to the large variability in the behavior of hM3Dq group. (p=0.2214, One-Way ANOVA followed by B and H-MC). Intriguingly, acute inhibition of mPFC-projecting vHIP neurons with the inhibitory DREADD hM4Di in *Mecp2* KO mice caused them to have a significant preference for the familiar mouse (n = 6, p=0.0004, One-sample t-test *Figure 5B* and *Figure 5—figure supplement 1A*) and show a difference in DI compared to mCherry expressing *Mecp2* KO mice whose DI was not different than chance (n = 10, p=0.8081, One-sample t-test; mCherry vs. hM4Di Mecp2 KO p=0.0197, One-Way ANOVA followed by B and H-MC). This effect was opposite to that of long-term inhibition of the vHIP-mPFC projection in

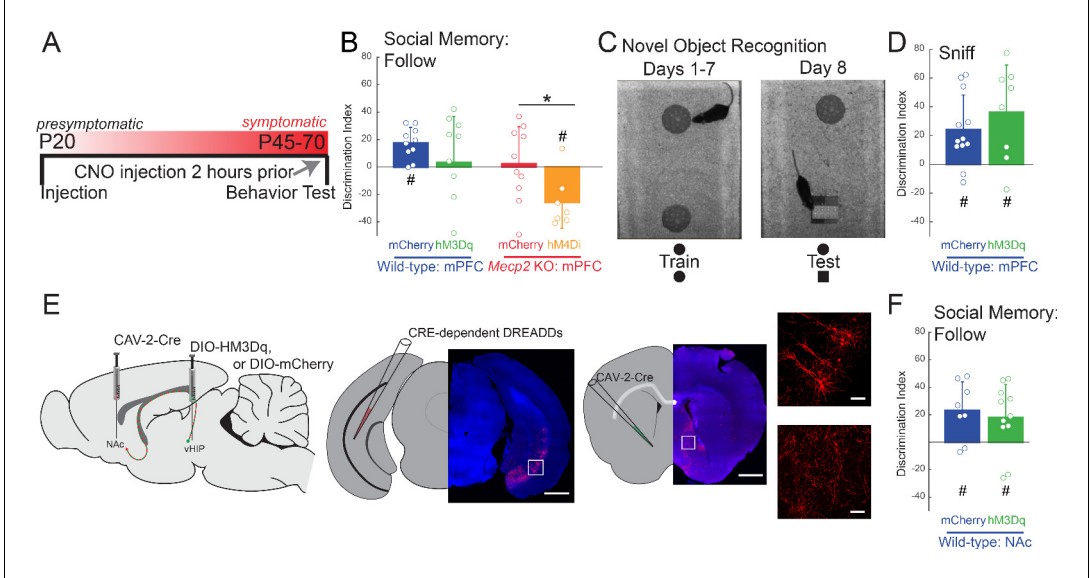

**Figure 5.** Acute manipulation of activity of vHIP-mPFC projection neurons regulates social memory in a task- and projection-specific manner. (A) Experimental timeline for acute DREADD manipulation of the vHIP-mPFC projection. (B) Time spent following either familiar or novel mice in unrestricted social interaction, scored by *JAABA* (n = 8 vHIP-mPFC mCherry WT mice, p=0.0135 Student's paired t-test; n = 10 vHIP-mPFC hM3Dq WT mice, p=0.6052 Student's paired t-test; n = 10 vHIP-mPFC mCherry *Mecp2* KO mice, p=0.9219 Wilcoxon paired test; n = 8 vHIP-mPFC hM4Di *Mecp2* KO mice, p=0.0469 Wilcoxon paired test). (C) Schematic of novel object recognition test. (D) Time spent sniffing either the familiar or novel object, scored by *JAABA* (n = 8 vHIP-mPFC mCherry WT mice, p=0.006 Student's paired t-test; n = 10 vHIP-mPFC hM3Dq WT mice, p=0.0266 Student's paired t-test). (E) Schematic of CAV2-Cre and DREADD injections to manipulate the vHIP-NAc projection. Injection sites show sparse mCherry labeling of vHIP neurons with identifiable axons in the NAc. Time spent following either familiar or novel mice in unrestricted social interaction, scored by *JAABA* (n = 10 vHIP-NAc mCherry WT mice, p=0.0391 Wilcoxon paired test; n = 11 vHIP-NAc hM3Dq WT mice, p=0.0189 Student's paired t-test). Mean ± SD; *p<0.05, **p<0.01. *Figure 5—source data 1*. See also *Figure 5—figure supplement 1*.
DOI: https://doi.org/10.7554/eLife.44182.028

The following source data and figure supplements are available for figure 5:

**Source data 1.** Acute manipulation of activity of vHIP-mPFC projection neurons regulates social memory in a task- and projection-specific manner.
DOI: https://doi.org/10.7554/eLife.44182.031

**Figure supplement 1.** Lack of effects of Acute DREADD stimulation by CNO on anxiety and general (non-social memory) behaviors.
DOI: https://doi.org/10.7554/eLife.44182.029

**Figure supplement 1—source data 1.** Acute manipulation of activity of vHIP-mPFC projection neurons regulates social memory in a task- and projection-specific manner.
DOI: https://doi.org/10.7554/eLife.44182.030

*Mecp2* KO mice (see *Figure 4C*) and the preference of naive untreated WT mice to follow the novel mouse (see *Figure 2G*). Despite these differences in social preference with WT mice, acute inhibition of the vHIP-mPFC projection caused *Mecp2* KO mice to display a significant preference in targeted social interactions, in contrast to control mCherry-expressing *Mecp2* KO mice.

We next determined whether the vHIP-mPFC projection encodes novelty in general, or specifically social novelty, by testing the acute effect of its activation on the novel object recognition test. WT mice expressing the excitatory DREADD hM3Dq in mPFC-projecting vHIP neurons showed the same discrimination for the novel object as control mCherry-expressing mice 2 hr after a single i.p. injection of CNO (n = 8 WT mCherry mice, p=0.0058; n = 10 hM3Dq, p=0.0172; One-sample t-test; mCherry vs. hM3Dq p=0.3524; Two-sample t-test; *Figure 5C–D* and *Figure 5—figure supplement 1B*), indicating that altering the activity of the long-range vHIP-mPFC projection does not affect hippocampal-dependent novel object recognition.

Do all vHIP projection neurons contribute to social memory, or just those projecting to the mPFC? To address this question, we injected CAV-2-Cre into the nucleus accumbens (NAc) and either control AAV8-DIO-mCherry or AAV8-DIO-hM3Dq into the vHIP of WT mice for selective excitation of NAc-projecting vHIP neurons (*Figure 5E*). Both hM3Dq-expressing and mCherry-expressing WT mice had a significant preference for the novel mouse compared to the familiar mouse and were not different from each other (n = 10 WT mCherry mice, p=0.00058; n = 11 WT hM3Dq mice, p=0.072; One-sample t-test; mCherry vs. hM3Dq p=0.6294; Two-sample t-test; *Figure 5F* and *Figure 5—figure supplement 1C*), indicating that mPFC-projecting vHIP neurons, but not NAc-projecting vHIP neurons, are necessary for the expression of social memory. Other than a small increase of walking time and aggressive behaviors in WT mice with hM3Dq expression in the vHIP-mPFC projection (Walk, p=0.0438, Student's t-test; Aggression, p=0.0214, Mann-Whitney Test due to nonparametric distributions), there were no differences in grooming behavior, locomotion, or anxiety-like behaviors between DREADD-expressing WT and *Mecp2* KO mice and their mCherry-expressing controls after CNO treatment (p>0.05; *Figure 5—figure supplement 1D–H*).

## Altered synaptic connectivity of long-range vHIP-mPFC projections in *Mecp2* KO mice

Differences in the spatiotemporal spread of VSD signals evoked by stimulation of the vHIP fiber bundle in mPFC slices between WT and *Mecp2* KO mice could reflect alterations in the innervation pattern of vHIP axons on different postsynaptic cell types in the mPFC. Morphological and electrophysiological recordings in vivo and in ex vivo slices from rats and WT mice have demonstrated that pyramidal neurons of the ventral CA1 and subiculum form monosynaptic connections with pyramidal neurons in layers 2/3 and 5, as well as with inhibitory interneurons in the PL and infralimbic (IL) regions of the mPFC (*Dembrow et al., 2015*; *Gabbott et al., 2002*; *Liu and Carter, 2018*; *Marek et al., 2018*). However, a quantitative analysis of the pattern of vHIP innervation onto different postsynaptic cell types was lacking. To identify the first-order postsynaptic neurons innervated by vHIP axons, we injected the trans-synaptic marker wheat germ agglutinin (WGA) into the vHIP (*Figure 6A*). After 30 hr to allow axonal and trans-synaptic transport in the mPFC (*Ruda and Coulter, 1982*), we performed immunohistochemistry for WGA and for the neuronal marker NeuN to account for potential WGA injection variability. We identified postsynaptic excitatory neurons by retrograde labeling and classified them based on their axonal projections as pyramidal tract (PT) neurons by injecting FluoroGold (FG) in the dorsal periaqueductal gray (dPAG), or as intratelencephalic (IT) neurons by injecting FG in the contralateral mPFC (c-mPFC) (*Figure 6B–D*). We identified postsynaptic inhibitory neurons by immunohistochemistry of the markers parvalbumin (PV), calretinin (CAL), and somatostatin (SOM). Regarding the distribution of WGA-positive neurons and neuronal subtypes across the different layers of the mPFC, there were no differences between WT and *Mecp2* KO mice for any of the cell types (*Figure 6—figure supplement 1*). In WT mice, the majority of WGA-labeled, NeuN-positive cells were projection pyramidal neurons, with 52% being IT neurons and 37% PT neurons, followed by 4% PV interneurons, 2% CAL interneurons, and 1% SOM interneurons (*Figure 6E*). The fraction of PT pyramidal neurons was significantly smaller in *Mecp2* KO mice (16%) (n = 9 sections from three mice for both WT and *Mecp2* KO mice, p=0.0128, Student's t-test), whereas the fraction of PV interneurons was significantly larger in *Mecp2* KO mice (9%) (p=0.0477, Student's t-test; *Figure 6F*). There were no significant differences in the fraction of IT pyramidal

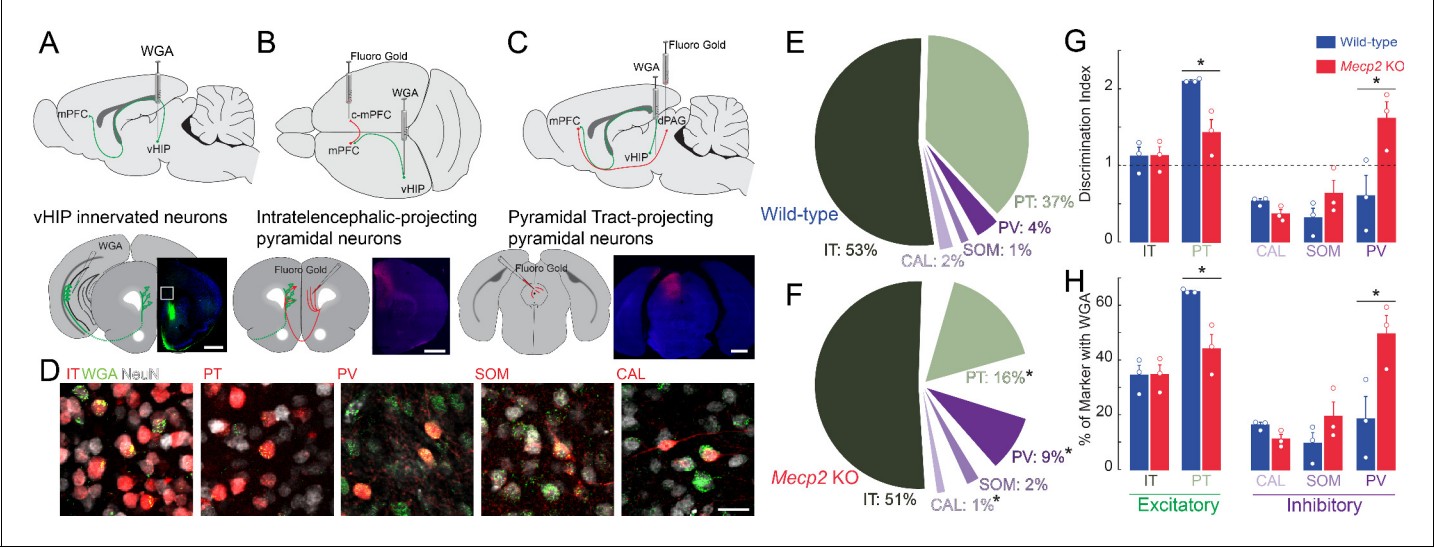

**Figure 6.** Trans-synaptic tracing of vHIP target neurons in the mPFC. (**A**) WGA injection sites for brains to be used for immunohistochemistry of interneuron markers. Scale bar = 1 mm. (**B**) Injection sites for brains to be used for identifying contralateral projecting mPFC neurons. Scale bar = 1 mm. (**C**) Injection sites for brains to be used for identifying dPAG projecting mPFC neurons. Scale bar = 1 mm. (**D**) Representative examples of WGA identification of inhibitory and excitatory neurons receiving vHIP innervation. Scale bar = 25 μm. (**E–F**) Breakdown of WGA innervated neurons by subtype in WT (**E**) and *Mecp2* KO mice (**F**) (IT p=0.8695; PT p=0.0128; CAL p=0.0621; SOM p=0.259; PV p=0.0477, Student's t-test). (**G**) Discrimination index of innervated cells, with at 'at chance' innervation being 1 (IT p=0.9646; PT p=0.0161; CAL p=0.591; SOM p=0.208; PV p=0.0425, Student's t-test). (**H**) Percent of neuron subtype receiving vHIP innervation (IT p=0.9646; PT p=0.0161; CAL p=0.591; SOM p=0.208; PV p=0.0425, Student's t-test) (n = 9 sections three mice for all groups). Mean ± SEM; *p<0.05, **p<0.01. *Figure 6—source data 1*. See also *Figure 6—figure supplement 1*.
DOI: https://doi.org/10.7554/eLife.44182.032

The following source data and figure supplement are available for figure 6:

**Source data 1.** Trans-synaptic tracing of vHIP target neurons in the mPFC.
DOI: https://doi.org/10.7554/eLife.44182.034

**Figure supplement 1.** Spatial distribution of different neuronal subtypes in the mPFC that are innervated by vHIP fibers in wild-type and *Mecp2* KO mice.
DOI: https://doi.org/10.7554/eLife.44182.033

neurons (51%), CAL interneurons, (1%), or SOM interneurons (2%) between *Mecp2* KO and WT mice (p=0.8695; p=0.0621; p=0.2590; Student's t-test; *Figure 6F*).

Although there is no evidence of neuronal cell death in RTT individuals and *Mecp2*-based mouse models (*Chen et al., 2001*), we accounted for potential differences in the density of different neuronal cell types by implementing a discrimination index for each mPFC neuron type that was trans-synaptically labeled with WGA. We gave an index of 1 when vHIP axons innervated neurons at chance values (having the same proportion of postsynaptic cell type in NeuN-positive and WGA-positive populations); by contrast, an index higher than one reflected innervation higher than chance and lower than one reflected innervation lower than chance. In the mPFC of WT mice, PT pyramidal neurons were preferentially innervated by vHIP axons (index = 2.1104, n = 9 sections from three mice, p=0.0001, one-sample Student's t-test; *Figure 6G*). IT pyramidal neurons and PV-positive interneurons did not have a significant discrimination index (index = 1.1192 and 0.6050, n = 9/3 mice each, p=0.4186 and p=0.2754, one-sample Student's t-test). Further, vHIP innervation of CAL and SOM interneurons occurred with a probability lower than chance (index = 0.5346 and 0.3165, n = 9/3 each mouse, p=0.0304 and p=0.0322, one-sample Student's t-test). By contrast, vHIP axons innervated PV cells in the mPFC of *Mepc2* KO mice more than in WT mice (n = 9/3, p=0.0425, Student's t-test), to the detriment of PT pyramidal cells (n = 9/3, p=0.0161, Student's t-test). We observed similar results when we assessed the percent of each neuronal subtype that received vHIP innervation; PV interneurons were innervated more at the expense of PT pyramidal cells in *Mecp2* KO compared to WT mice (*Figure 6H*). Combined, these results indicate that the pattern of innervation of vHIP axons in the mPFC changes from mainly targeting excitatory projection pyramidal neurons in WT

mice to preferentially targeting PV-expressing inhibitory GABAergic interneurons (*Figure 6G and H*).

## Enhanced vHIP-mPFC synaptic strength in *Mecp2* KO mice

Larger peak VSD signals evoked by stimulation of the vHIP fiber bundle in mPFC slices from *Mecp2* KO mice could reflect either more or stronger excitatory synapses between presynaptic vHIP axons and postsynaptic mPFC neurons. We estimated the numerical density and size of *en passant* presynaptic terminals along afferent axons within the mPFC by labeling them with mCherry delivered by AAV2 injected into either the vHIP or the c-mPFC, and then performing immunohistochemistry of the presynaptic vesicle marker VGLUT1 (*Figure 7A–E*). We performed automated detection and size analysis using *Bouton Analyzer* (*Gala et al., 2017*) (*Figure 7E*). The numerical densities of mCherry-expressing *en passant* presynaptic terminals belonging to vHIP neurons and those belonging to c-mPFC neurons were comparable between WT and *Mecp2* KO mice across all cortical layers of the mPFC (p>0.05, Two-way ANOVA followed by B and H-MC; *Figure 7F* and *Figure 7—figure supplement 1*). However, the weighted size of individual presynaptic boutons of vHIP axons in layer 5 of the mPFC was significantly larger in *Mecp2* KO mice (n = 993 boutons in WT mice, n = 664 *Mecp2*

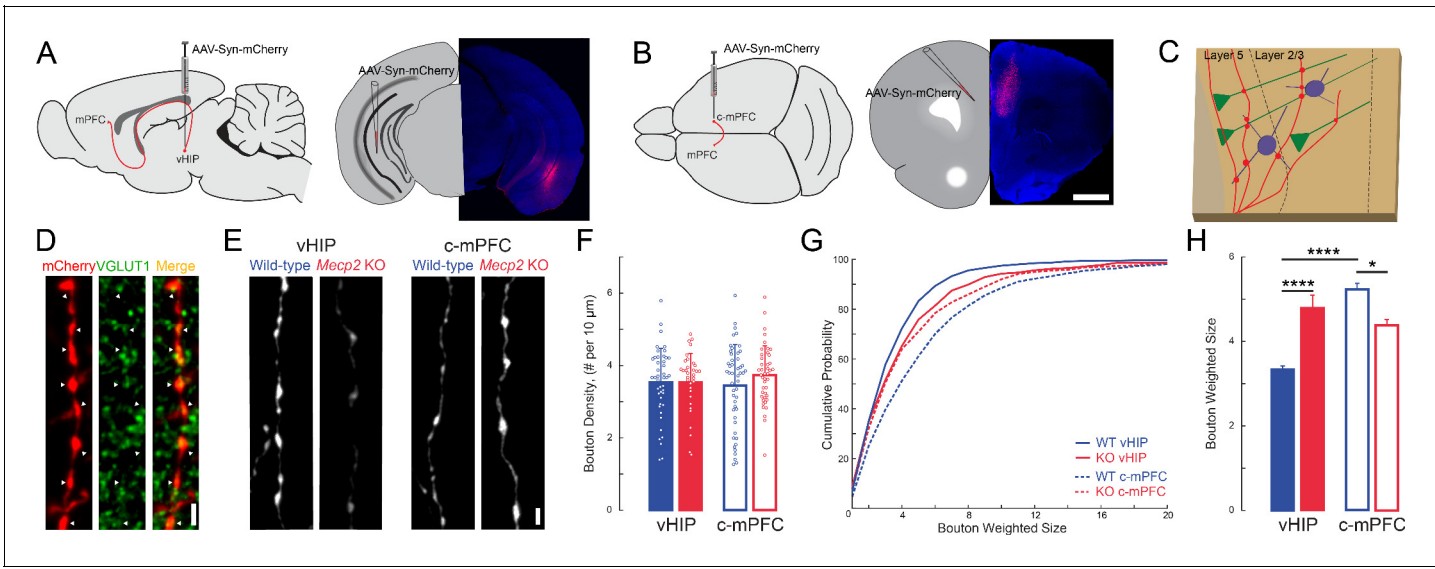

**Figure 7.** The size of presynaptic boutons is altered in layer 5 of the mPFC of *Mecp2* KO mice. (**A–B**) Schematic and representative examples of AAV2-hSyn-mCherry injection sites for identifying vHIP axons (**A**) or contralateral mPFC axons (**B**). Scale bars 1 mm. (**C**) Schematic of vHIP axons in the mPFC. (**D**) Axonal swellings identified as presynaptic boutons co-labeled for mCherry and VGLUT1. Scale bar 2 μm. (**E**) Representative examples of mCherry-filled presynaptic boutons. Scale bar 2 μm. (**F**) Numerical densities of axonal boutons per length of vHIP and c-mPFC axons located in layer 5 of mPFC (n = 45 axons WT vHIP; n = 36 *Mecp2* KO vHIP; n = 49 WT c-mPFC; n = 43 KO c-mPFC; WT vHIP vs. *Mecp2* KO vHIP, p=0.9925; WT vHIP vs. WT c-mPFC, p=0.8317 *Mecp2* KO vHIP vs. *Mecp2* KO c-mPFC, p=0.6765; WT c-mPFC vs. *Mecp2* KO c-mPFC, p=0.6765; Interaction p=0.0135; Axon p=0.9592; Genotype p=0.3102; Two-way ANOVA with Benjamini and Hochberg multiple comparisons). Mean ± SD. (**G**) Cumulative probability distributions of the estimated size of presynaptic boutons comparing vHIP and c-mPFC axons in mPFC layer 5 of WT and *Mecp2* KO mice. (**H**) Average weighted bouton sizes (vHIP WT mice vs. *Mecp2* KO mice, p<0.0001; c-mPFC WT mice vs. *Mecp2* KO mice, p=2320; vHIP vs. c-mPFC WT mice, p<0.0001; vHIP vs. c-mPFC *Mecp2* KO mice, p=0.0133; n = 993 WT vHIP boutons; n = 682 KO vHIP; n = 792 WT c-mPFC; n = 577 KO c-mPFC; Interaction p=0.1193; Axon p=0.3135; Genotype p=0.2967; Two-way ANOVA with Benjamini and Hochberg multiple comparisons). Mean ± SEM; *p<0.05, **p<0.01. *Figure 7—source data 1*. See also *Figure 7—figure supplement 1*.

DOI: https://doi.org/10.7554/eLife.44182.035

The following source data and figure supplements are available for figure 7:

**Source data 1.** The size of presynaptic boutons is altered in layer 5 of the mPFC of *Mecp2*KO mice.

DOI: https://doi.org/10.7554/eLife.44182.038

**Figure supplement 1.** Presynaptic boutons are not altered in Layer 2/3 of the mPFC in *Mecp2* KO mice.

DOI: https://doi.org/10.7554/eLife.44182.036

**Figure supplement 1—source data 1.** The size of presynaptic boutons is altered in layer 5 of the mPFC of *Mecp2*KO mice.

DOI: https://doi.org/10.7554/eLife.44182.037

KO mice, p<0.0001, Two-way ANOVA followed by B and H-MC), though there was no difference in layer 2/3 (n = 543 boutons in WT mice, n = 720 in *Mecp2* KO mice, p=2565, Two-way ANOVA followed by B and H-MC; *Figure 7G–H* and *Figure 7—figure supplement 1C–D*). In contrast, the sizes of presynaptic boutons of c-mPFC axons in layers five were significantly smaller in *Mecp2* KO mice (n = 782 WT mice, n = 575 *Mecp2* KO mice, p=0.0133; Two-way ANOVA followed by B and H-MC, *Figure 7G–H*), though not in layer 2/3 (n = 698 WT, n = 818 *Mecp2* KO, p=8422, Two-way ANOVA followed by B and H-MC, *Figure 7—figure supplement 1C–D*). Interestingly, presynaptic boutons of vHIP axons in layer 5 of the mPFC were significantly smaller than those of c-mPFC axons in WT mice, a difference absent in *Mecp2* KO mice due to larger vHIP boutons and smaller c-mPFC boutons (WT, p<0.0001; *Mecp2* KO 0.2320; *Figure 7G–H*). These results are reminiscent of the amplitude of VSD signals in ex vivo mPFC slices evoked by either vHIP or intracortical stimulation, which were biased towards intracortical stimulation in WT mice, and of comparable amplitude in *Mecp2* KO mice due to both larger vHIP-evoked VSD signals and smaller intracortical-evoked responses (see *Figure 3L*).

Because the size of presynaptic terminals is positively correlated with the area of the presynaptic active zone and the number of docked synaptic vesicle (*Harris and Sultan, 1995*; *Murthy et al., 2001*), which in turn is positively correlated with the area of the postsynaptic density and the volume of dendritic spines (*Harris and Stevens, 1989*; *Harris and Weinberg, 2012*), as well as their content of α-amino-3-hydroxy-5-methyl-4-isoxazolepropionic acid receptors (AMPAR) (*Matsuzaki et al., 2004*), we hypothesized that larger VSD responses to vHIP fiber stimulation in the mPFC of *Mecp2* KO mice reflect higher synaptic strength. To selectively stimulate different axonal projections onto the same postsynaptic mPFC neuron during whole-cell intracellular recordings, we used two light-sensitive opsins with shifted excitation spectra (*Klapoetke et al., 2014*). We injected AAV2s expressing the red-shifted opsin Chrimson into the ipsilateral vHIP, and those expressing the blue-shifted opsin Chronos into the c-mPFC (*Figure 8A–B* and *Figure 8—figure supplement 1A–B*). In mice expressing Chrimson in the vHIP and Chronos in the c-mPFC, a brief (1–4 ms) pulse of either red (630 nm) or blue (430 nm) light evoked monotonic inward currents in layer five neurons in the presence of 4-AP (100 mM), TTX (1 μM), and 4 mM $Ca^{2+}$, which represent monosynaptic excitatory postsynaptic currents (EPSCs) (*Petreanu et al., 2009*). The amplitude of red light vHIP-evoked EPSCs was significantly larger in pyramidal neurons from *Mecp2* KO mice compared to those from WT mice at all ranges of light pulse durations (n = 17 cells in 6 slices from 6 WT mice; n = 11/5/5 *Mecp2* KO mice; p=0.0118; Two-way RM ANOVA; *Figure 8C*), whereas the amplitude of vHIP-evoked monosynaptic EPSCs in layer five interneurons (identified by their size, shape, input resistance, and capacitance; see *Figure 8—figure supplement 1C–D*) was comparable in both genotypes (n = 11 cells in 7 slices from 7 WT mice; n = 12/9/9 *Mecp2* KO mice; p=0.4327; Two-way RM ANOVA; *Figure 8D*). In contrast, blue light stimulation of Chronos-expressing c-mPFC axons evoked EPSCs of comparable amplitude in both layer five pyramidal neurons and interneurons from WT and *Mecp2* KO mice at all ranges of light pulse durations (p=0.1617; p=0.6129; Two-way RM ANOVA; *Figure 8E and F*). The cell-by-cell normalization of the amplitude of EPSCs evoked by red light stimulation of Chrimson-expressing vHIP axons to the amplitude of EPSCs evoked by blue light stimulation of Chronos-expressing c-mPFC axons revealed that vHIP-evoked EPSCs were larger in layer five pyramidal from *Mecp2* KO mice compared to WT mice (103% vs. 54%; p=0.00083; Two-way RM ANOVA; *Figure 8G*). In contrast, normalized EPSC amplitudes in interneurons were not significantly different between *Mecp2* KO and WT mice (97% vs. 81%; p=0.3208; Two-way RM ANOVA) (*Figure 8H*). These results are reminiscent of the normalized amplitude of VSD signals evoked by stimulation of the fluorescently labeled vHIP fiber bundle compared to intracortical stimulation (see *Figure 3L*).

As a direct measure of postsynaptic strength, we calculated the ratio of the AMPAR component of the EPSC (recorded at −70 mV) to that of their NMDAR component (recorded at +40 mV) in the same neuron (*Figure 8I*). The AMPAR/NMDAR ratio of EPSCs evoked by red light vHIP stimulation was larger in layer five pyramidal neurons from *Mecp2* KO mice compared to those from WT mice (n = 11 cells from six slices from WT mice; n = 13/5/5 *Mecp2* KO mice; p=0.0114; Two-way ANOVA followed by B and H-MC), whereas the AMPAR/NMDAR ratio of EPSCs evoked by blue light c-mPFC stimulation was smaller in *Mecp2* KO mice compared to WT mice (p=0.0114; Two-way ANOVA followed by B and H-MC; *Figure 8J*). In contrast, the AMPAR/NMDAR ratio of EPSCs evoked by either vHIP or c-mPFC stimulation in interneurons was not significantly different between *Mecp2* KO and WT mice (n = 9/7/7 WT mice; n = 11/9/9 *Mecp2* KO mice; p=0.8843 vHIP

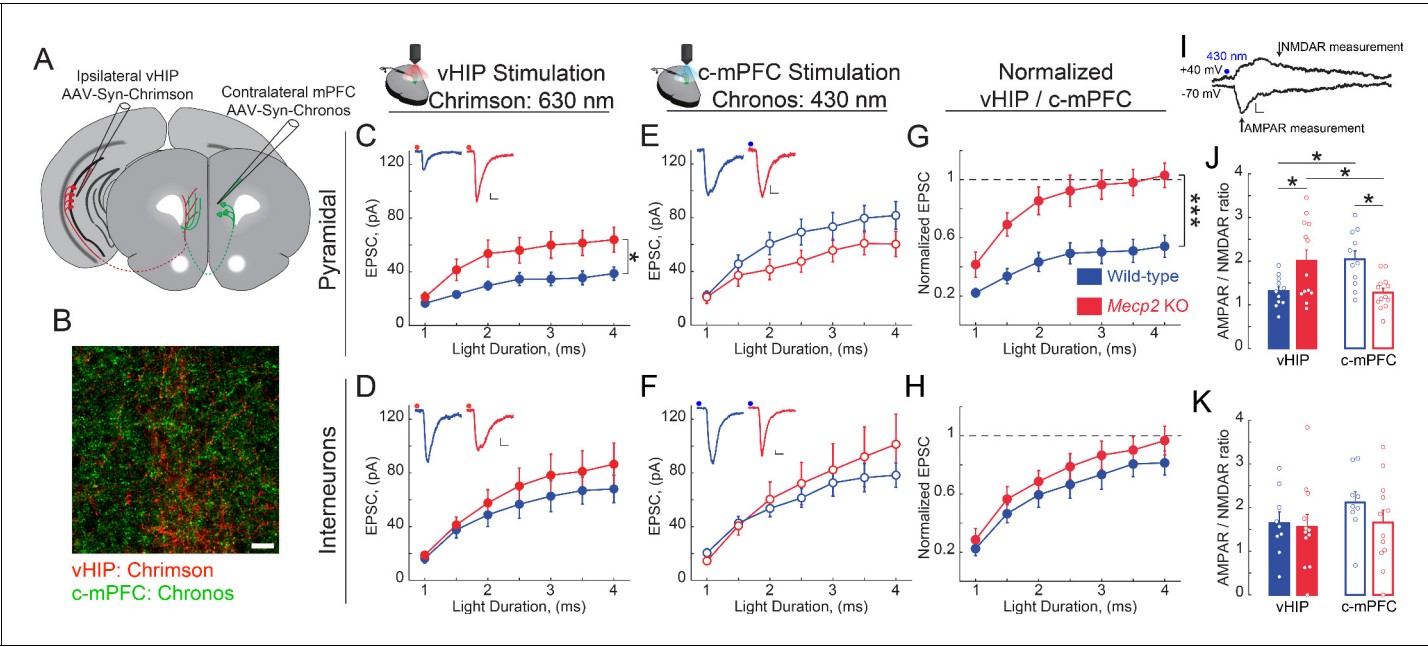

**Figure 8.** vHIP synapses on mPFC layer five pyramidal neurons are stronger in *Mecp2* KO mice. (**A**) Schematic of injection sites for Chrimson in the ipsilateral vHIP and Chronos in the contralateral mPFC. (**B**) Representative image of an mPFC slice with Chrimson-expressing vHIP afferents and Chronos-expressing c-mPFC afferents. Scale bar 50 µm. (**C and D**) Input-output relationship of vHIP afferent (red light) evoked responses in pyramidal neurons (**C**) Interaction p=0.0093; Stim p<0.0001; Genotype p=0.0118, Two-way RM ANOVA) and interneurons (**D**) Interaction p=0.2692; Stim p<0.0001; Genotype p=0.4327, Two-way RM ANOVA) with representative traces (inset). (**E and F**) Input-output relationship of c-mPFC afferent (blue light) evoked responses in pyramidal neurons (**E**) Interaction p=0.69643; Stim p<0.0001; Genotype p=0.1617, Two-way RM ANOVA) and interneurons (**F**) Interaction p=0.4222; Stim p<0.0001; Genotype p=0.6129, Two-way RM ANOVA) with representative traces (inset) (Scale bars 10 pA/12 ms). (**G and H**) The amplitude of vHIP afferent (red light)-evoked EPSCs was normalized to the peak EPSCs evoked by blue light stimulation of c-mPFC afferents in pyramidal neurons (**G**) Interaction p=0.0098; Stim p<0.0001; Genotype p=0.0003, Two-way RM ANOVA) and interneurons (**H**) Interaction p=0.9644; Stim p<0.0001; Genotype p=0.3208, Two-way RM ANOVA) (n = 17 cells from 6 slices from six mice WT pyramidal neurons, 11/7/7 WT interneurons, 11/5 KO pyramidal cells, 12/9/9 KO interneurons). (**I**) Representative example trace showing the time windows were measurements were made to calculate the AMPAR/NMDAR ratio (Scale bar 5 pA/10 ms). (**J and K**) AMPAR/NMDAR ratios of vHIP afferent (red light)-evoked and c-mPFC afferent (blue light)-evoked responses in pyramidal cells (**J**) WT vs. *Mecp2* KO vHIP, p=0.0114, WT vs. *Mecp2* KO c-mPFC, p=0.0038; WT vHIP vs. WT c-mPFC, p=0.0114; *Mecp2* KO vHIP vs. *Mecp2* KO c-mPFC, p=0.0114; Interaction p=0.0002; Input p0.9912; Genotype p=0.8544; Two-way ANOVA with Benjamini and Hochberg multiple comparisons) and interneurons (**K**) WT vs. *Mecp2* KO vHIP, p=0.8843; WT vs. *Mecp2* KO c-mPFC, p=0.8121; WT vHIP vs. WT c-mPFC, p=0.8121; *Mecp2* KO vHIP vs. *Mecp2* KO c-mPFC, p=0.8843; Interaction p=0.4896; Input p=0.2825; Genotype p=0.6265; Two-way ANOVA with Benjamini and Hochberg multiple comparisons) (n = 11 cells from 6 slices from six mice WT pyramidal cells; 9/7/7 WT interneurons; 13/5/5 *Mecp2* KO pyramidal cells for both vHIP and c-mPFC stimulation; 11/9/9 *Mecp2* KO interneurons). Mean ± SEM; *p<0.05, **p<0.01. *Figure 8—source data 1*. See also *Figure 8—figure supplement 1*.

DOI: https://doi.org/10.7554/eLife.44182.039

The following source data and figure supplements are available for figure 8:

**Source data 1.** vHIP synapses on mPFC layerfivepyramidal neurons are stronger in *Mecp2*KO mice.
DOI: https://doi.org/10.7554/eLife.44182.042

**Figure supplement 1.** Specific activation of axons expressing opsins with non-overlapping light sensitivity.
DOI: https://doi.org/10.7554/eLife.44182.040

**Figure supplement 1—source data 1.** vHIP synapses on mPFC layerfivepyramidal neurons are stronger in *Mecp2*KO mice.
DOI: https://doi.org/10.7554/eLife.44182.041

stimulation; p=0.8121 c-mPFC stimulation; Two-way ANOVA followed by B and H-MC; *Figure 8K*). Combined, these results demonstrate that vHIP excitatory synapses are selectively stronger on layer five pyramidal neurons, but not interneurons, in the mPFC of *Mecp2* KO mice.

## Discussion

Here, we characterized the projection from the vHIP to the PL region of the mPFC at the structural and functional levels in WT mice, and described its atypical features in the *Mecp2* KO model of Rett

syndrome. Because the vHIP has been implicated in social memory, and due to the involvement of the mPFC PL subregion in sociability and social novelty encoding, we tested the role of the vHIP projection to the mPFC in social behaviors. By chemogenetically manipulating neuronal activity selectively in mPFC-projecting vHIP CA1 neurons, we demonstrated that these projection neurons regulate social memory in a specific and selective manner, only influencing the discrimination between social targets without affecting other aspects of social interactions.

Coherent, synchronous oscillations between the vHIP and the mPFC underlie working memory tasks in rats (*Gordon, 2011*). Entrainment of these oscillations occurs in the first postnatal week (*Brockmann et al., 2013*) and is driven by monosynaptic glutamatergic projections from pyramidal neurons of the ventral CA1 and subiculum to pyramidal neurons and GABAergic interneurons in the IL and PL subregions of the mPFC (*Anastasiades et al., 2018*; *Dégenètais et al., 2003*; *Dembrow et al., 2010*; *Thierry et al., 2000*). Despite the wealth of information linking this projection to neuropsychiatric disorders (*Li et al., 2015*), little is known about the synaptic and cellular bases of this long-range projection. To better define the connectivity of this circuit, we performed trans-synaptic tracing to identify postsynaptic targets of vHIP afferents and determined how this innervation pattern is altered in *Mecp2* KO mice. Although recent evidence suggests that monosynaptic excitatory vHIP inputs to the mPFC are strongest on pyramidal neurons in layer 2/3 of the IL, but dominated by feed-forward inhibition onto them (*Marek et al., 2018*), there is also evidence of strong monosynaptic excitatory innervation of IT-projecting pyramidal neurons in layer 5 of the PL (*Liu and Carter, 2018*). Our trans-synaptic tracing data support the latter observations, with over 50% of WGA-positive neurons being IT-projecting pyramidal neurons. We also identified PV-positive interneurons as the inhibitory subgroup most innervated by vHIP axons, which is consistent with the observation of vHIP-driven feed-forward inhibition in the IL (*Marek et al., 2018*). In *Mecp2* KO mice, there are fewer vHIP-innervated PT-projecting pyramidal neurons, which have been shown to encode social dominance (*Franklin et al., 2017*). This suggests a basis for the impairments in social memory performance observed in *Mecp2* KO mice. However, there are more vHIP-innervated PV-positive interneurons, which may lead to tonic inhibition of the mPFC network due to the hyperactivity of the vHIP in *Mecp2* KO mice (*Calfa et al., 2011*). Interestingly, the density of *en passant* presynaptic boutons along vHIP axons in the mPFC was not altered in *Mecp2* KO mice, indicating a redistribution of excitatory vHIP inputs on different cell types in the mPFC, which results in an atypical wiring pattern.

Although the number of vHIP boutons in the mPFC was not altered, their individual volume was larger in layer 5 of the PL region of the mPFC in *Mecp2* KO mice. By contrast, vHIP boutons in layer 2/3 had a more prominent bimodal distribution of volumes compared to WT mice. Because the size of presynaptic boutons is correlated with synaptic strength (*Murthy et al., 2001*), we tested the strength of vHIP-mPFC synapses onto both pyramidal neurons and interneurons. EPSCs evoked by optogenetic excitation of vHIP fibers in layer 5 mPFC pyramidal cells were larger and had higher AMPAR/NMDAR ratios in *Mecp2* KO mice compared to WT littermates. Interestingly, vHIP-evoked EPSC amplitudes and AMPAR/NMDAR ratios in interneurons were not affected in *Mecp2* KO mice. Together with the trans-synaptic identification of postsynaptic targets of vHIP terminals in the mPFC, these results indicate that vHIP axons in *Mecp2* KO mice innervate fewer pyramidal neurons with stronger synaptic strength, but they have similar innervation strength onto more inhibitory interneurons. Such altered connectivity is reflected in the pattern of vHIP-evoked neuronal depolarizations in mPFC slices revealed by high-speed voltage imaging. In these studies, the larger amplitudes reflect stronger excitatory synapses onto pyramidal neurons, while the spatiotemporal spread throughout the mPFC slice and over time reflects both an altered connectivity pattern and feed-forward inhibition. It is possible that a larger proportion of inhibitory neurons is chronically activated by hyperactive vHIP inputs in *Mecp2* KO mice, causing tonic inhibition of the mPFC network, and that vHIP activation of IT-projecting pyramidal neurons overcomes this inhibition.

LTP at vHIP-mPFC synapses is impaired in *Mecp2* KO mice (*Figure 2—figure supplement 2*). Together with larger vHIP-evoked VSD signals (*Figure 2*) and whole-cell intracellular recordings from layer five pyramidal neurons (*Figure 8*), these data are similar to those observed in hippocampal CA3-CA1 synapses of *Mecp2* KO mice. There, naive excitatory synapses in *Mecp2* KO mice have several features of potentiated synapses due to impaired synaptic GluA1 trafficking, suggesting a saturation of the dynamic range available for LTP (*Li et al., 2016*). A similar mechanism prevents homeostatic synaptic plasticity in *Mecp2* KO hippocampal neurons in culture (*Xu and Pozzo-Miller,*

2017). Current work is aimed at characterizing the mechanism(s) of impaired LTP at vHIP-mPFC synapses in *Mecp2* KO mice.

Deficits in the E/I balance within the mPFC have been linked to impaired sociability in WT mice, as well as in mouse models of ASDs (*Selimbeyoglu et al., 2017*; *Yizhar et al., 2011*; *Brumback et al., 2018*). In addition, the hippocampal network is integral to the expression of social memory (*Hitti and Siegelbaum, 2014*; *Meira et al., 2018*; *Okuyama et al., 2016*). Because network activity within the vHIP and the mPFC, as well as the projection pattern of vHIP afferents in the mPFC, are altered in *Mecp2* KO mice, we characterized their sociability, social interaction, and social memory. We performed these experiments using computer vision to track multiple freely interacting mice and a computer learning algorithm trained to identify different behaviors (*Kabra et al., 2013*; *Ohayon et al., 2013*; *Robie et al., 2017*). This automatic and unbiased screen of social interactions revealed that, despite showing typical sociability, *Mecp2* KO mice displayed an atypical behavior of jumping at other mice more than following or sniffing, which was not associated with aggression. This finding, to our knowledge, is the first description of an atypical social interaction in a mouse model of ASD. We also identified a deficit of social memory in *Mecp2* KO mice, which failed to discriminate between a co-housed littermate and a novel mouse as the target of their social interactions under unrestricted conditions.

To demonstrate a causal role of altered vHIP-mPFC inputs on atypical social behaviors in *Mecp2* KO mice, we mimicked the characteristic hippocampal hyperactivity of *Mecp2* KO mice in WT mice by long-term chemogenetic activation with the excitatory hM3Dq DREADD selectively expressed in mPFC-projecting vHIP neurons. Such long-term excitation impaired social memory, without affecting other types of social interactions. Chronic inhibition of vHIP-mPFC projecting neurons in WT mice also impaired social memory, indicating that there is a set level of proper neuronal activity in this long-range projection, with any deviation resulting in social memory deficits. Even though the vHIP-mPFC projection has been causally tied to anxiety-like behaviors (*Padilla-Coreano et al., 2016*), long-term chemogenetic manipulation of vHIP-mPFC projection neurons had no major consequences on the time spent in the center of the arena, the time spent engaging in grooming, or the overall locomotor behavior, compared to CNO-treated WT mice expressing Cre-driven mCherry in mPFC-projecting vHIP neurons. This finding demonstrates the selectivity of this manipulation of the vHIP-mPFC projection to social memory. Voltage imaging of network responses in mPFC slices from hM3Dq-expressing mice revealed stronger vHIP inputs and, surprisingly, weaker responses evoked by stimulation of intracortical inputs (which were not chemogenitically manipulated), resembling the responses observed in *Mecp2* KO mice. These results may reflect feed-forward homeostatic mechanisms within the mPFC microcircuit, as has been hypothesized to give rise to the dichotomy in the direction of E/I imbalances between limbic and cortical structures in models of neuropsychiatric diseases (*Nelson and Valakh, 2015*).

Despite the fact that many brain regions are dysfunctional in *Mecp2* KO mice, selective long-term inhibition of vHIP-mPFC projection neurons was sufficient to improve their social memory. In addition, social memory discrimination scores were negatively correlated with the amplitude of vHIP-driven depolarizations in mPFC slices: larger voltage dye signals corresponded to worse social memory performance, whereas smaller voltage dye signals corresponded with better expression of social memory. By contrast, long-term inhibition of vHIP-mPFC projection neurons did not affect voltage dye responses evoked by intracortical stimulation, and these responses did not correlate with social memory performance.

Acute manipulation of vHIP-mPFC projection neurons immediately prior to the behavioral test also affected social memory: their excitation with hM3Dq caused a discrimination index not different than chance, whereas inhibition with hM4Di improved social discrimination in *Mecp2* KO mice. These data suggest that the vHIP-mPFC projection is necessary for the recall of social memory, as opposed to its maintenance or initial formation. Because acute manipulation impaired social memory in WT mice, we used this paradigm to test the specificity of the vHIP-mPFC projection to the social aspect of memory, as well as its selectivity by testing another vHIP projection target. Acute excitation of hM3Dq-expressing mPFC-projecting vHIP neurons did not affect novel object recognition in WT mice, indication that this manipulation did not affect overall hippocampal function. Furthermore, regulation of social memory was selective to mPFC-projecting vHIP neurons because excitation of hM3Dq-expressing vHIP neurons projecting to the NAc did not affect social memory, which is at odds with a previous report concluding that vHIP-NAc projection is required for social memory

(**Okuyama et al., 2016**). A key difference between these studies is our intersectional viral approach to selectively and exclusively express DREADDs in specific projection neurons of the vHIP.

In summary, we demonstrate that the vHIP-mPFC projection regulates social memory in WT mice, and that its dysfunction causes social memory deficits in *Mecp2* KO mice. Defining the synaptic bases of social behaviors provides insight and potential targets for therapies in psychiatric disorders associated with vHIP-mPFC dysfunction, such as autism and schizophrenia.

# Materials and methods

**Key resources table**

| Reagent type (species) or resource | Designation | Source or reference | Identifiers | Additional information |
|---|---|---|---|---|
| Strain, strain background (Mus musculus) | Mecp2$^{tm1.1Jae}$ | Mutant Mouse Resource and Research Center | MMRRC:000415-UCD | |
| Genetic reagent (Virus) | AAV-hSyn-Chronos-GFP | UNC Vector Core | UNC Vector Core | AAV2 |
| Genetic reagent (Virus) | AAV-hSyn-Chrimson R-tdTomato | UNC Vector Core | UNC Vector Core | AAV2 |
| Genetic reagent (Virus) | AAV-hSyn-mCherry | UNC Vector Core | UNC Vector Core | AAV2 |
| Genetic reagent (Virus) | CAV-2-Cre-GFP | CNRS Biocampus Montpellier | CNRS Biocampus Montpellier | |
| Genetic reagent (Virus) | pAAV8-DIO-hSyn-HM4D(Gi)-mCherry | Addgene | 44362 | AAV8 |
| Genetic reagent (Virus) | pAAV-DIO-hSyn-HM3D(Gq)-mCherry | Addgene | 44361 | AAV8 |
| Genetic reagent (Virus) | pAAV-DIO-hSyn-mCherry | Addgene | 50459 | AAV8 |
| Antibody | Anti-mCherry (Rabbit polyclonal) | Abcam | ab167453 | (1:500) |
| Antibody | Anti-vGlut1 (Guinea Pig polyclonal) | Synaptic Systems | 135304 | (1:500) |
| Antibody | Anti-GFP (chicken polyclonal) | Abcam | ab13970 | (1:500) |
| Antibody | Anti-PV (Rabbit polyclonal) | Abcam | ab11427 | (1:500) |
| Antibody | Anti-PV (Guinea Pig polyclonal) | Synaptic Systems | 195004 | (1:750) |
| Antibody | Anti-cFos (Guinea Pig polyclonal) | Synaptic Systems | 226004 | (1:500) |
| Antibody | Anti-CAL (Chicken polyclonal) | Synaptic Systems | 214106 | (1:750) |
| Antibody | Anti-SST (Guinea Pig polyclonal) | Synaptic Systems | 366004 | (1:750) |
| Antibody | Anti-NeuN (Mouse monoclonal) | Synaptic Systems | 266011 | (1:500) |
| Antibody | Anti-NeuN (Guinea Pig polyclonal | Synaptic Systems | 266004 | (1:1000) |
| Antibody | Anti-Lectin (Rabbit polyclonal) | Sigma | T4144-1VL | (1:2000) |
| Antibody | Alexa Fluor 488 Goat Anti-Rabbit | Jackson Immuno Research | 111-545-003 | (1:500) |

*Continued on next page*

*Continued*

| Reagent type (species) or resource | Designation | Source or reference | Identifiers | Additional information |
|---|---|---|---|---|
| Antibody | Alexa Fluor 594 Goat Anti-Rabbit | Jackson Immuno Research | 111-585-003 | (1:500) |
| Antibody | Alexa Fluor 594 Goat Anti-Guinea Pig | Jackson Immuno Research | 106-585-003 | (1:500) |
| Antibody | Alexa Fluor 594 Goat Anti-Chicken | Jackson Immuno Research | 103-585-155 | (1:500) |
| Antibody | Alexa Fluor 647 Goat Anti-Mouse | Jackson Immuno Research | 115-605-003 | (1:500) |
| Antibody | Biotinylated Goat Anti-Rabbit | Vector Laboratories | BA-1000 | (1:200) |
| Antibody | Biotinylated Goat Anti-Guinea Pig | Vector Laboratories | BA-7000 | (1:200) |
| Antibody | Streptavidin, Alexa Fluor 405 | Thermo Fisher Scientific | S-32351 | (1:1500) |
| Antibody | Streptavidin, Alexa Fluor 488 | Thermo Fisher Scientific | S-32354 | (1:1500) |
| Antibody | Streptavidin, Alexa Fluor 594 | Thermo Fisher Scientific | S-32356 | (1:1500) |
| Chemical compound, drug | Dextran-Alexa-594 10,000 MW | Thermo Fisher Scientific | D22913 | |
| Chemical compound, drug | Wheat Germ Agglutinin | Vector Laboratories | L-1020 | |
| Chemical compound, drug | FluoroGold | Fluorochrome | | |
| Chemical compound, drug | RetroBeads | Lumafluor Inc | | |
| Chemical compound, drug | Tetrodotoxin | Alomone Labs | | |
| Chemical compound, drug | 4-Aminopyridine | Sigma | 275875 | |
| Chemical compound, drug | D-AP5 | Tocris | 106 | |
| Software, algorithm | Motr | Janelia research center | | https://github.com/motr/motr |
| Software, algorithm | JAABA | Janelia research center | | https://github.com/kristinbranson/JAABA |
| Software, algorithm | Novel Object Recognition Add-on for JAABA | This paper | | https://github.com/PhillipsML/NOR |
| Software, algorithm | Voltage Sensitive Dye Analysis Code | This paper | | https://github.com/PhillipsML/VoltageDyeAnalysis |
| Software, algorithm | GraphPad Prism, version 8 | GraphPad | RRID:SCR_002798 | |
| Software, algorithm | TIWB | DOI: 10.1093/jmicro/dfy015. | | |
| Software, algorithm | Neuroplex | Red Shirt Imaging | | |
| Software, algorithm | ImageJ, FIJI | DOI: 10.1038/nmeth.2019 | | |
| Software, algorithm | Matlab, 2015b & 2017b | Mathworks | | |
| Other | Born Blonde Maxi | Clairol | | |

## Contact for reagent and resource sharing

Further information and requests for resources and reagents should be directed to and will be fulfilled by the Lead Contact, Dr. Lucas Pozzo-Miller (lucaspm@uab.edu).

## Experimental model and subject details

Female mice with deletions of exon three in the *Mecp2* gene (Mecp2$^{tm1.1Jae}$; *Chen et al., 2001*) were obtained from the Mutant Mouse Regional Resource Center (University of California, Davis), and maintained in a pure C57/BL6 background by crossing them with male WT C57/BL6 mice. All experimental subjects were male hemizygous Mecp2$^{tm1.1Jae}$ mice, referred to as *Mecp2* KO mice. Subjects classified as presymptomatic were tested between P20 and P24. Subjects that exhibited Rett-like symptoms, such as hypoactivity, hind limb clasping, resting tremors, and reflex impairments (*Guy et al., 2001*), were classified as symptomatic (between P45-P60). Age-matched male WT littermates were used as controls. Mice were handled and housed according to the Committee on Laboratory Animal Resources of the National Institutes of Health. All experimental protocols were reviewed and approved annually by the Institutional Animals Care and Use Committee of the University of Alabama at Birmingham.

## Method details

### Intracranial injections

Mice were anesthetized with 4% isoflurane vapor in 100% oxygen gas and maintained with 1–2.5% isoflurane vapor in 100% oxygen gas mixtures. Mice were aligned in a stereotactic frame (Kopf Instruments, Tujunga, CA), and their body temperature was measured with a rectal probe and maintained with a heating pad. A midline incision was made down the scalp, and a dental drill used to perform a small craniotomy. A 2.5 µL syringe (Hamilton Company, Reno, NV) was used to inject solutions (viruses, RetroBeads, fluorescent dextran, FluoroGold, or WGA) at a rate of 0.25 µL/min using a microsyringe pump (UMP3 UltraMicroPump, Micro4; World Precision Instruments, Sarasota, FL). The needle was slowly extracted from the injection site over 10 min, after which the incision was closed with surgical glue. All ages and coordinates for each experiment are relative to the bregma and listed as anterior/posterior (A/P), medial/lateral (M/L), and dorsal/ventral (D/V).

### Immunohistochemistry

Mice were anesthetized with an i.p. injection of ketamine (100 mg/kg) and transcardially perfused with ice-cold 1X phosphate-buffered saline (PBS), followed by ice-cold 4% paraformaldehyde (PFA) in 1X PBS. The brain was dissected and postfixed in 4% PFA overnight. Brains were sectioned at 30 µm thickness with a vibratome (PELCO 100, model 3000; Ted Pella Inc, Redding, CA) and stored at 4°C in 1X PBS containing 0.01% sodium azide. Free-floating sections were permeabilized using 0.25% Triton-100X for 15 min and subsequently incubated in blocking solution (0.01% sodium azide, 2% bovine serum albumin (BSA), 0.1% Triton-100X, 2M glycine, and 10% goat serum in 1X PBS) for 1 hr. Antibody diluent consisted of 0.01% sodium azide, 2% BSA, 0.1% Triton-100X, and 5% goat serum in 1X PBS. Primary antibodies were diluted in antibody diluent at concentrations listed below and incubated for 36 hr at room temperature. After washing 3 times for 5 min in 1X PBS, secondary antibodies were diluted in antibody diluent and incubated for 4 hr at room temperature. Sections were washed 3 times for 5 min in 1X PBS before mounting with Vectashield mounting media (Vector Biolabs, Malvern, PA).

### Amplification

For primary antibodies requiring further amplification (c-Fos, mCherry, and GFP), a biotinylation step was added following the primary antibody incubation. Sections were incubated in biotinylated anti-host of the primary antibody (Vector Biolabs, Malvern, PA) at a concentration of 1:200 in antibody diluent for 2 hr. After washing 3 times for 5 min in 1X PBS, sections were incubated in streptavidin-conjugated fluorophore (Alexa-405, Alexa-488, Alexa-594; Thermo Fisher Scientific, Waltham, MA) diluted in antibody diluent at a concentration of 1:1600 for 4 hr. Sections were washed 3 times for 5 min before mounting with Vectashield mounting media (Vector Biolabs, Malvern, PA).

## Injection of WGA

For identification of mPFC neurons innervated by the vHIP, P45 mice received a 500 nL injection of 4% WGA (Vector Laboratories, Burlingame, CA) (*Ruda and Coulter, 1982*) into the vHIP (3.8 A/P, 3.3 M/L, 3.5 D/V from the bregma). Mice were sacrificed 30 hr later, a time point that our pilot tests confirmed the transsynaptic transfer to the 1 st order neurons, and not to other neurons down the synaptic chain. To avoid cross-reactivity between the anti-WGA primary antibody and other antibodies, sections were first incubated with anti-WGA and then underwent subsequent biotinylation and streptavidin steps. Following the last wash, sections were again blocked for 1 hr. Immunohistochemitry for neuronal subtypes and NeuN was subsequently performed following the standard protocol.

## Injection of FluoroGold

For identification of mPFC pyramidal neurons based on their projections, P30 mice received a 250 nL injection of 2% FluoroGold (Fluorochrome, Denver, CO) (*Schmued and Fallon, 1986*) either in the right mPFC (1.45 A/P, 0.5 M/L, 1.45 D/V from the bregma) or in the dPAG (1.18 A/P, 4.2 M/L, 2.36 D/V from the bregma with a needle angle of 26˚). At P45, the same mice were injected with 500 nL of 4% WGA into the vHIP (3.8 A/P, 3.3 M/L, 3.5 D/V from the bregma) and sacrificed 30 hr later.

## Identification of socially-activated neurons

For identification of vHIP projection neurons, P35 mice received injections of red RetroBeads (Lumafluor, Durham, NC) (*Quattrochi et al., 1989*) in the LH (1.34 A/P, 1.1 M/L, 5.3 D/V from the bregma) and green RetroBeads in the mPFC (1.45 A/P, 0.5 M/L, 1.45 D/V from the bregma). Two weeks after surgery, mice underwent 3 days of testing acclimation (3 min of handling and 10 min inside the 16 × 10 in test box. On day 4, mice were placed in the test box containing either a littermate sentinel (social condition) or a toy mouse (object condition) and allowed to interact for 10 min before being returned to the home cage. After 1 hr, the same test mouse was placed back in the same box, which now included either a novel sentinel mouse (social condition) or a novel toy mouse (object condition), and allowed to interact for 10 min before being returned to the home cage. Previous reports indicate the vHIP neurons are activated by social encounter regardless of novelty (*Okuyama et al., 2016*). We therefore chose to present a strong stimulus to recruit as many vHIP neurons as possible by using sequential interactions with both a littermate and novel mouse. Test mice were anesthetized with ketamine (100 mg/kg, i.p.) 45 min after the last interaction and perfused for subsequent immunostaining. vHIP sections from mice containing RetroBeads were stained for the immediate early gene c-Fos and NeuN. mPFC sections from non-surgical mice were stained for c-Fos. All sections were processed for immunohistochemistry at the same time, and images were taken consecutively using identical imaging settings in a confocal microscope. For quantification of c-Fos fluorescence intensity in RetroBead labeled neurons, the soma of NeuN-positive cells co-labeled with RetroBeads were manually circled using FIJI (ImageJ, NIH). The measurement was then redirected to the c-Fos channel and fluorescence intensity calculated. All RetroBead labeled neurons contained within the ventral subiculum and ventral CA1 of 6 stained sections each from four mice were measured. Sections which had atypically high background fluorescence were excluded; the remaining were normalized to background fluorescence (determined by measuring the intensity of NeuN negative region). Because all RetroBead-labeled neurons were labeled, we did not set a threshold for considering a neuron c-Fos positive and instead report the fluorescence intensity for each individual neuron in a cumulative frequency distribution (*Figure 1D*) and the intensity average per mouse (*Figure 1E*). Statistical tests were conducted on averaged data. For c-Fos positive cells in the mPFC, FIJI was used to make images binary and automatically detect particles and measure fluorescence. Identical settings were used during this process. Because this experiment necessitated the use of a threshold for binarization and therefore not all neurons were measured, we used a fluorescence intensity cutoff of 80 arbitrary units to define cells as c-Fos positive. These data were reported statistically tested as number of c-Fos positive neurons per section, averaged by mouse (*Figure 1—figure supplement 1*).

## Imaging and quantification of presynaptic terminals

P20 mice received injections of AAV2-Syn-mCherry (UNC Vector Core, Chapel Hill, NC) either in the vHIP (3.8 A/P, 3.3 M/L, 3.5 D/V from the bregma; 500 nL) or in the right mPFC (1.45 A/P, 0.5 M/L,

1.45 D/V from the bregma; 250 nL). They were perfused 4 weeks later, and 30 μm sections were cut from the mPFC and stained with anti-mCherry antibodies. All images were acquired using a 63X (1.4 NA) oil immersion objective in an LSM-800 Airyscan confocal microscope (Zeiss, Oberkochen, Germany) using identical settings (laser power, pinhole, photomultiplier tube, current, gain, and offset). Axons were semi-manually traced using *NCTracer* in FIJI (*Longair et al., 2011*), and boutons were quantified using *BoutonAnalyzer* (*Gala et al., 2017*) running in Matlab (MathWorks, Natick, MA). Statistical tests were conducted on density per axon and size per bouton.

## Electrophysiology

### Ex vivo brain slices

Mice were anesthetized with ketamine (100 mg/kg, i.p.) and transcardially perfused with an ice-cold modified 'cutting' artificial cerebrospinal fluid (aCSF) containing 87 mM NaCl, 2.5 mM KCl, 0.5 mM $CaCl_2$, 7 mM $MgCl_2$, 1.25 mM $NaH_2PO_4$, 25 mM $NaHCO_3$, 25 mM glucose, and 75 mM sucrose, bubbled with 95% $O_2$/5% $CO_2$. Coronal sections (300 μm thick) of the mPFC cut at an angle of 10° from the coronal plane were prepared using a vibratome (VT1200S; Leica Biosystems, Wetzlar, Germany), transferred to a submerged recovery chamber filled with 'recording' aCSF (125 mM NaCl, 2.5 mM KCl, 2 mM $CaCl_2$, 1 mM $MgCl_2$, 1.25 mM $NaH_2PO_4$, 25 mM $NaHCO_3$, and 25 mM glucose, bubbled with 95% $O_2$/5% $CO_2$), kept at 30°C for 30 min, and then allowed to recover for 1 hr at room temperature before use.

### High-speed imaging of voltage-sensitive dye signals

For visualization of vHIP afferent fibers in mPFC slices, P30 mice received bilateral injections of 0.4 μL of Dextran-Alexa-594 (10,000 MW; Thermo Fisher Scientific, Waltham, MA) in the vHIP, at 3.88 A/P, 3.3 M/L, 3.5 D/V from the bregma. After 2 weeks, mice were sacrificed and individual ex vivo slices were stained with 30 μM of the voltage-sensitive fluorescent dye RH414 (Anaspec, Fremont, CA) in aCSF for 1 hr at room temperature, and transferred to an immersion chamber continuously perfused (2 mL/min) with 'recording' aCSF saturated with 95%$O_2$/5% $CO_2$ and kept at 30°C. A theta glass electrode filled with 'recording' aCSF was positioned within the fluorescently labeled vHIP fiber bundle, and another theta glass electrode was placed in line with the cortical column across from the vHIP fiber electrode. In some experiments, an extracellular electrode filled with 'recording' aCSF (1–3 MΩ) was positioned in layer 2/3 to record fEPSPs with an Axoclamp amplifier (Molecular Devices, San Jose, CA) in current-clamp mode; signals were sampled at 2 kHz, amplified 10 times with a pre-amplifier (Model 210; Brownlee Precision, now NeuroPhase, Palo Alto, CA), digitized at 10 kHz (ITC-18; InstruTech, Longmont, CO), and acquired with custom-written software (TI WorkBench) (*Inoue, 2018*) in a MacMini computer (Apple, Cupertino, CA). RH414 was excited at 530 ± 50 nm with a phosphor-pumped LED (Heliophor; 89 North, Williston, VT), and its filtered fluorescence (535 ± 50 nm band-pass, 580 nm beam-splitter, 594 nm long-pass; Semrock, Rochester, NY) was imaged in an inverted microscope (IX71; Olympus, Tokyo, Japan) through a 10 × 0.5 NA objective (Fluar; Zeiss, Oberkochen, Germany) and acquired with a scientific CMOS camera running at 2500 frames-per-second at full 128 × 128 pixel resolution (NeuroCMOS-SM128; RedShirt Imaging, Decatur, GA), controlled by NeuroPlex software (RedShirt Imaging, Decatur, GA) in a Windows computer (Dell, Round Rock, TX); the electrophysiology computer was synchronized with the imaging computer by TTL pulses. The input-output relationship of the amplitude of VSD signals was obtained by delivering 12 different stimulus intensities with 30 μA increments. The input-output relationship of VSD amplitudes matched that of the the slope of fEPSPs, and VSD signals followed the kinetics of individual fEPSPs (*Figure 3—figure supplement 1*). LTP of VSD signals and fEPSPs was induced by high-frequency stimulation of vHIP afferent fibers, which consisted of trains of pulses at 300 Hz for 0.5 s repeated 10 times with 3 min intervals (*Huang et al., 2004*). Induction of LTP in mPFC slices from symptomatic *Mecp2* KO mice required partial antagonism of $GABA_A$ receptors with 5 μM picrotoxin to reduce the hyperactivity driven by vHIP stimulation; LTP induction in mPFC slices from presymptomatic *Mecp2* KO mice only required 1 μM picrotoxin.

VSD signals were analyzed using custom written Matlab codes (https://github.com/PhillipsML/VoltageDyeAnalysis; *Phillips, 2019b*, copy archived at https://github.com/elifesciences-publications/VoltageDyeAnalysis), as following: VSD responses in layer 2/3 were determined in a semi-automated manner with boundaries based on the percent of cortical thickness; the boundary between layers 5

and 2/3 was 20% of the cortical thickness from the white matter, and the boundary between layers 2/3 and 1 was 10% of the cortical thickness from the pial surface. Maximum responses were reported as the mean dF/F from a 3 × 3 pixel Region Of Interest (ROI) placed within layer 2/3 with the highest response in the 6 ms following stimulation, which corresponds to the peak of the fEPSPs. The spatio-temporal spread was calculated as the area under the curve (AOC) of the spread in the cortical area (% of total) and time (ms). For the correlation between VSD signals and social memory performance, only one slice per mouse was used (the slice with the median response). Statistical tests were performed on slices as the replicate.

## Intracellular whole-cell recordings

For optogenetic activation of vHIP and contralateral mPFC afferents, a total of 0.5 µL of AAV2-Syn-Chrimson-tdTomato (UNC Vector Core, Chapel Hill, NC) was injected into the left vHIP (250 nL at 3.5 A/P, 3.3 M/L, 3.5 D/V from the bregma and 250 nL at 3.1 A/P, 3.3 M/L, 3.0 D/V from the bregma) and 250 nL of AAV2-Syn-Chronos-GFP (UNC Vector Core, Chapel Hill, NC) into the right mPFC (1.45 A/P, 0.5 M/L, 1.45 D/V from the bregma). After allowing 4 weeks for opsin expression, mPFC ex vivo slices were transferred to an immersion chamber continuously perfused (2 mL/min) with 'recording' aCSF (120 mM NaCl, 2.5 mM KCl, 2 mM $CaCl_2$, 1 mM $MgCl_2$, 25 mM $NaHCO_3$, 1.4 mM $NaH_2PO_4$, 21 mM glucose, 0.4 mM Na-ascorbate, and 2 mM Na-pyruvate bubbled with 95% $O_2$/5% $CO_2$) and kept at 30˚C. Addition of 1 µM TTX and 100 mM 4-AP, and increasing $Ca^{2+}$ to 4 mM ensured recordings of monosynaptic responses (*Petreanu et al., 2009*). Whole-cell recording electrodes were pulled from borosilicate glass (World Precision Instruments, Sarasota, FL) and filled with 120 mM Cs-gluconate, 17.5 mM CsCl, 10 mM Na-HEPES, 4 mM Mg-ATP, 10 mM $NA_2$-creatine phosphate, and 0.2 mM Na-EGTA; this yielded a resistance of 4 ± 0.5 MΩ in aCSF. Whole-cell currents were recorded in voltage-clamp mode with an Axopatch-200B amplifier (Molecular Devices, San Jose, CA), filtered at 2 kHz, digitized at 10 kHz (ITC-18; InstruTech, Longmont, CO), and acquired with TI WorkBench in a G5 PowerMac computer (Apple, Cupertino, CA). Opsins were excited with monochromatic light from a Laser-LED illumination system (Lumen Dynamics, now Excelitas Technologies, Waltham, MA) attached to the epifluorescence port of a DM-LFS fixed-stage upright microscope (Leica Biosystems, Wetzlar, Germany), and focused onto the slice through a Zeiss 63X (1.0 NA) water immersion lens (Zeiss, Oberkochen, Germany), which was also used for visualized whole-cell recordings under infrared-differential interference contrast. TI WorkBench (*Inoue, 2018*) controlled and synchronized the Laser-LED (Lumen Dynamics, now Excelitas Technologies, Waltham, MA) stimulation with electrophysiological recordings and fluorescence imaging with an electron multiplying charge-coupled device (QuantEM 512SC; Photometrics, Tucson, AZ). Chronos was excited with 430 nm light, and Chrimson was excited with 630 nm light. In control experiments, 630 nm light evoked inward currents in layer five pyramidal neurons in slices expressing only Chrimson in vHIP axons, but not those expressing Chronos in c-mPFC axons (*Figure 8—figure supplement 1A–B*). Conversely, 430 nm light of an intensity that evoked inward currents in slices expressing only Chronos in c-mPFC axons, did not evoke any responses in slices expressing only Chrimson in vHIP axons (*Figure 8—figure supplement 1B*). Neurons were classified as pyramidal neurons or interneurons based on their morphology, input resistance, and whole-cell capacitance (estimated from the exponential decay of the current response to a test voltage step) (*Figure 8—figure supplement 1c-d*). Neurons whose input resistance changed more than 20% during the recordings or had unclamped spikes (i.e., action currents) were excluded. Statistical tests considered neuron the biological replicate.

## Behavioral assays

### Three-chamber social assay

Mice were acclimated to being handled during 3 min each for 3 days prior to the testing day at the same time as testing. All handling and testing were done in the dark phase of the 12 hr light/12 hr dark cycle, with the experimenter wearing a red headlamp and infrared illumination for digital videography. Mice were placed in the center chamber of a three-chambered box that contained empty inverted pencil cups in the two side chambers, and were allowed to freely explore. After 5 min of acclimation, mice were shepherded back to the center chamber, and blocks were put in place over the side openings. A novel mouse was put under one of the pencil cups on one of the two sides,

with the side being interleaved between trials. The blocks were lifted and the test mouse allowed to freely explore the chambers for 10 min. After this time, the test mouse was again shepherded into the middle compartment and blocked there. A second novel mouse was placed under the previously empty pencil cup. The side blocks were removed and the test mouse was allowed to freely explore for another 10 min. After testing each mouse, the apparatus was thoroughly cleaned with isopropanol. Test mice spending more than 75% of the acclimation time in one compartment were removed from the study. The amount of time the test mouse spent actively sniffing each pencil cup (either empty or containing sentinel mice) was quantified and the discrimination index was calculated as [(Time investigating Mouse Cup - Time investigating Empty Cup) / (Time investigating Mouse Cup + Time investigating Empty Cup) * 100] for the sociability test and [(Time investigating Novel Mouse Cup - Time investigating Familiar Mouse Cup) / (Time investigating Novel Mouse Cup + Time investigating Familiar Mouse Cup) * 100] for the memory test. Statistical tests were preformed on the discrimination indices per mouse.

## Unrestricted social Assay

One week prior to testing, the back of sentinel mice was dyed with blond hair dye (Born Blonde Maxi, Clairol) with differing patterns for tracking by computer vision. Mice were acclimated to handling and to the testing box containing clean bedding for 3 days prior to the testing day. All acclimation and testing (3 min handling, 10 min in the testing box) were done in the dark phase with experimenter wearing a red headlamp, and video acquisition using infrared illumination. At the beginning of the testing day, sentinel mice from different cages were placed in a neutral cage to acclimate to one another. The test mice were placed in the testing box and were videotaped for 10 min. Then, sentinel mice (one cage-mate of the test mouse and an unknown from a different cage) were placed in the testing box with the test mouse, and allowed to freely interact while being videotaped for 10 min. After this time, the sentinel mice were placed back in the neutral cage and the test mouse returned to the home cage. The test box was cleaned and filled with new bedding between each test mouse. Each sentinel mice interacted with a maximum of 5 test mice, and were discarded if they fought with other sentinels or were excessively grooming. After all mice had been tested, sentinel mice were individually videotaped for 10 min for computer training. Individual and test videos were fed to the *Motr* program (https://github.com/motr/motr; *Ohayon et al., 2013*) to create tracks that were sent to *JAABA* (https://github.com/kristinbranson/JAABA; *Kabra et al., 2013*; *Robie et al., 2017*; *Branson, 2019*) for unbiased computer identification of behaviors. *JAABA* classifiers were trained on pilot data sets. Behavioral scores for social memory assessment were taken from the first 4 min of the trial (*Figure 2*), and separated based on the mouse target (cage-mate or novel). The rationale of using the first 4 min of the total 10 min of the unrestricted social interaction assay is based on the observation that more than 75% of the social interactions occur during the first 4 min. After that time, there is no longer a difference in time spent following the target mice compared to the time spent exploring the environment, as the test mouse becomes acclimated to their presence. As a result of this, the Discrimination Index after the first 4 min become a less reliable estimate preference in the social interactions (*Figure 2—figure supplement 2*). Behavioral scores for social memory and other behaviors were taken from the entire video, as different behaviors emerged at later times during the trial, and social behavior times pooled between novel and cage-mate mouse. Memory discrepancy scores were calculated as (Time Following Novel - Time Following Familiar) / (Time Following Novel + Time Following Familiar) * 100. Mice not interacting with sentinels for more than 3 s (out of 240) were excluded. Preference for the novel was considered a value greater than 0. Statistical tests were performed on the discrimination indices per mouse for memory, or for all behaviors, time per behavior for each mouse.

## Novel object recognition

Mice were acclimated to handling and to the testing box containing clean bedding and two identical objects for 7 days prior to the testing day (handled for 3 min, placed in the testing box for 10 min). On the 8th day, mice were returned to the testing chamber where one of the objects had been replaced with a novel object. Time spent interacting with each object was scored using a custom code addition (https://github.com/PhillipsML/NOR; *Phillips, 2019a*, copy archived at https://github.com/elifesciences-publications/NOR)        to        *JAABA*        (https://github.com/kristinbranson/

JAABA; *Branson, 2019*), which was taught to identify active sniffing behavior, and allow quantification of the sniffing time within a user-defined region of interest around each object. Memory scores were calculated for the first 4 min of the test day. Memory discrepancy scores were calculated as (Time Following Novel - Time Following Familiar) / (Time Following Novel + Time Following Familiar) * 100. Statistical tests were performed on the discrimination indices per mouse.

## Chemogenetic manipulation with DREADDs and CNO

### Long-term treatment

P20 WT and *Mecp2* KO mice were injected with CAV-2-Cre (Biocampus, Institute of Molecular Genetics, Montpellier, France) into the mPFC; 250 nL at 1.45 AP, 0.5 ML, and 1.45 DV, and with either AAV8-hSyn-DIO-mCherry, AAV8-hSyn-DIO-hM4Di(Gi)-mCherry, or AAV8-hSyn-DIO-hM3Dq (Gq)-mCherry into the vHIP; 500 nL at 3.5 AP, 3.3 ML, and 4.0 DV (all viruses from the UNC Vector core). CNO (Tocris Bioscience) was first dissolved in DMSO (5 mg in 200 μL), and then diluted in 200 mL of water with 5 mM saccharine (Sigma Aldrich) (*Carvalho Poyraz et al., 2016*). Mice were allowed *ad libitum* access to this solution in a standard drinking bottle starting at P34. Mice were tested using the unrestricted social interaction assay at P45. Mice were kept on CNO and sacrificed 3–5 days after behavioral testing for the preparation of ex vivo mPFC slices for voltage-sensitive dye imaging. Previous reports have validated the efficacy of long-term activity modulation using CNO activation of DREADDs, though all potential adaptations were not tested (*Cheng et al., 2019*).

### Acute treatment

P20 WT mice were injected with CAV-2-Cre either into the mPFC; 250 nL at 1.45 AP, 0.5 ML, and 1.45 DV; or the NAc; 250 nL at 0.9 AP, 0.9 ML, and 3.8 DV; and AAV8-hSyn-DIO-mCherry or AAV8-hSyn-DIO-hM3Dq(Gq)-mCherry into the vHIP; 500 nL at 3.5 AP, 3.3 ML, and 4.0 DV. P20 *Mecp2* KO mice were injected with CAV-2-Cre into the mPFC; 250 nL at 1.45 AP, 0.5 ML, and 1.45 DV; and AAV8-hSyn-DIO-mCherry or AAV8-hSyn-DIO-hM4Di(Gi)-mCherry into the vHIP; 500 nL at 3.5 AP, 3.3 ML, and 4.0 DV. Mice were acclimated to IP needle pokes for 2 weeks prior to behavioral tests, though were not injected with any substance. On testing day, mice received a single IP injection of CNO (3 mg per kg of body weight) 2 hr before testing (*Smith et al., 2015*).

## Quantification and statistical analysis

All experiments were performed blinded to genotype, with the exception of AAV injections expressing DREADDs. Because DREADD-mediated long-term excitation of the already hyperactive vHIP of *Mecp2* KO mice will not reveal new information neither on Rett syndrome nor typical social behaviors, we did not include that experimental group to reduce the number of experimental mice. Thus, the experimenters were not blinded to genotype when injecting mice in order to optimize the number of animals to be used in these experiments. The experimenters were blinded during behavioral testing, though some *Mecp2* KO mice were easy to identify due to their Rett-like symptoms. The implementation of computer vision to score all behavioral data, and of Matlab codes to analyze all VSD data were additional steps taken to ensure unbiased acquisition and data analyses. In addition, all statistical analyses were performed blinded to the genotype and treatment groups using Prism (GraphPad) and Matlab. Power analyses were performed using G*Power (*Faul et al., 2009*). Statistical tests used in each experiment are provided in main text, within associated figure legend, or within the statistical table in supplemental information. Data first underwent a test for normalcy: either D'Agostino and Pearson's or, in the case that the number of replicates was not sufficient for this test, Shapiro-Wilk normalcy test. The decision to use either parametric or non-parametric tests were dependent on this outcome. ANOVAs were conducted in tandem with Benjamani and Hochberg Multiple Comparisons, unless otherwise stated. Memory discrepancy indices were first tested using t-tests (two groups) or ANOVAs (more than two groups) to determine differences between groups and secondly tested in a one-sample t-test against chance value (0) to determine if the preference was significant. Analysis of input-output relationships used Two-Way Repeated Measures (RM) ANOVA. Significance was defined as $p < 0.05$, with the specific statistical test provided in main text or within associated figure legend in addition to the statistical table in supplemental information. Sample sizes (n) refer to number of cells, number of slices, or number of animals, with the specific convention provided in the main text or within the associated figure legend. Significance

conventions are as follows: *: p<0.05; **: p<0.01; ***: p<0.001. Sample sizes are provided in main text, within associated figure legend, or within the statistical table in supplemental information. Behavioral data are presented as mean ± SD, all other data are presented as mean ± SEM.

## Acknowledgements

This research was supported by NIH grant R56-NS103089-01A1 (LP-M), the Civitan International Foundation (MLP), NIH training grant T32-NS061788-04 (MLP), and *Rettsyndrome.org* (LP-M). We thank Dr. Wei Li for comments on the manuscript, Ms. Karen Ayala-Baylon for mouse colony mainte- nance, and Dr. Takafumi Inoue (Waseda University, Tokyo, Japan) for electrophysiology and imaging software. We thank the Biostatistics Clinic of UAB's Center for Clinical and Translational Science for assistance with statistical analyses. We also thank Mr. Rahul Gaini, Ms. Allison Fusilier, and Ms. Melissa Bentley for their contributions to data acquisition and analyses in early phases of this project.

## Additional information

### Funding

| Funder | Grant reference number | Author |
|---|---|---|
| National Institutes of Health | R56-NS103089-01A1 | Lucas Pozzo-Miller |
| Civitan International | Civitan Emerging Scholar: W John Rynearson Award | Mary L Phillips |
| International Rett Syndrome Foundation | | Lucas Pozzo-Miller |
| National Institutes of Health | T32-NS061788-04 | Mary L Phillips |

The funders had no role in study design, data collection and interpretation, or the decision to submit the work for publication.

### Author contributions

Mary L Phillips, Conceptualization, Data curation, Software, Formal analysis, Supervision, Funding acquisition, Validation, Investigation, Visualization, Methodology, Writing—original draft, Writing— review and editing; Holly Anne Robinson, Software, Formal analysis, Investigation, Writing—review and editing; Lucas Pozzo-Miller, Conceptualization, Resources, Supervision, Funding acquisition, Writing—review and editing

### Author ORCIDs

Mary L Phillips https://orcid.org/0000-0003-4696-1555
Lucas Pozzo-Miller https://orcid.org/0000-0001-6085-5527

### Ethics

Animal experimentation: Mice were handled and housed according to the Committee on Laboratory Animal Resources of the National Institutes of Health. All experimental protocols were reviewed and approved annually by the Institutional Animals Care and Use Committee of the University of Ala- bama at Birmingham (IACUC-20114).

### Decision letter and Author response

Decision letter https://doi.org/10.7554/eLife.44182.045
Author response https://doi.org/10.7554/eLife.44182.046

## Additional files

### Supplementary files

• Transparent reporting form
DOI: https://doi.org/10.7554/eLife.44182.043

### Data availability

All data generated or analyzed during this study are included in the manuscript and supporting files. Custom code used is available at https://github.com/PhillipsML/VoltageDyeAnalysis and https://github.com/PhillipsML/NOR (copies archived at https://github.com/elifesciences-publications/VoltageDyeAnalysis and https://github.com/elifesciences-publications/NOR).

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
