## [Decision Letter]

Thank you for submitting your article "Ventral hippocampal projections to the medial prefrontal cortex regulate social memory" for consideration by *eLife*. Your article has been reviewed by 3 peer reviewers, one of whom is a member of our Board of Reviewing Editors, and the evaluation has been overseen by Laura Colgin as the Senior Editor. The following individuals involved in review of your submission have agreed to reveal his identity: Jeffrey L Neul (Reviewer #3).

The reviewers have discussed the reviews with one another, and the Reviewing Editor has drafted this decision to help you prepare a revised submission.

Summary:

This study uses a large array of techniques to examine the properties of vHIP – mPFC connections in wild type and MeCP2KO animal to establish the role of this circuit in regulation of social behavior. Although a large number of synaptic and cellular deficits have been characterized in RTT models, the circuit deficits associated with this disorder remain poorly understood. Therefore, the reviewers overall agree that the current study is in principle an important addition to existing studies on RTT pathophysiology. However, the authors need to revise several aspects of their paper in order to validate the conclusions. More rigorous data analyses (in particular statistical tests) are needed to strengthen this rather extensive study.

Essential revisions:

1) As stated in the individual reviewers' comments below, statistical analysis in each figure should strengthened. In particular the authors should present better justifications for the statistical tests they used and also validate the sample sizes (i.e., perform power analyses). If all claims hold up under statistical scrutiny, then many of the other criticisms in the individual reviews become minor.

2) The authors should comment on the potential caveats of their experimental design in several places (use of cFOS signals and long term use of DREADD-CNO system). Experimental design details require better justification overall.

3) Related to the above point, the design of the social interaction tests requires justification. It is unclear why they authors chose to only assess the first 4 minutes of the 10 minute encounter for the behavioral test. A reasonable rationale should be provided to explain this choice.

The individual reviews are included below for your information.

*Reviewer #1:*

This study uses a large array of techniques to examine the properties of vHIP – mPFC connections in wild type and MeCP2KO animal to establish the role of this circuit in regulation of social behavior. Although a large number of synaptic and cellular deficits have been characterized in RTT models, the circuit deficits associated with this disorder remain poorly understood. Therefore, the current study is in principle a welcome addition to existing canon of studies on RTT pathophysiology. However, the authors need to clarify several aspects of their experiments and provide more technical details to make this rather extensive study more accessible to a wider audience.

1) Figure 1: cFOS staining should use histograms to better present the distribution of fluorescence intensities? What do these intensities present, cells, clusters? Clarify the relationship between neuronal activity and cFOS staining. Why count neurons, rather than quantify staining intensity per neuron?

2) Figure 3: The relationship between fEPSPs and voltage sensitive dye signals. EPSP slope is a more reliable reporter for synaptic responses, whereas fEPSP amplitudes are rather non-linear reporters for postsynaptic activity. What do the authors think they are monitoring using voltage sensitive dyes (synaptic responses, firing or a loosely defined measure of "activity")?

3) The authors should be commended for the well-controlled DREADD-CNO experiments. That said, the authors primarily perform long term experiments using this system. It is important to validate the DREADD system works as expected under the conditions of long-term (11 days?) exposure to CNO. It is plausible to expect some level of adaptation and plasticity in the system. However, the authors' interpretation of their result does not take this possibility into account.

4) It is extremely difficult to interpret the experiments presented in Figure 8 in terms of a change in presynaptic efficacy. The authors use an extremely artificial setting (4-AP + TTX+ 4 mM Ca^2+^) to evaluate synaptic efficacy via optogenetic stimulation. It is difficult ascertain if the differences seen in these experiments are indeed due to bigger boutons and increased presynaptic release probability.

*Reviewer #2:*

The authors demonstrate that the projection from ventral hippocampus to mPFC is hyperactive in symptomatic Rett mice, yet activity driven by intracortical stimulation is decreased. The electrophysiological findings in Figures 3 and Figure 4, and anatomical findings in Figure 6, are convincing and compelling. However, the link to behavior is tenuous, largely because of issues with the statistics. It is difficult to gauge the claims made in Figure 1, Figure 2, Figure 4, Figure 5. Although this is a potentially very interesting manuscript, the statistical issues make it difficult to assess the impact of the paper, as well as what additional experiments or analyses may be needed.

Specifically:

Figure 1 represents an extremely ambitious experiment, as the authors compare c-Fos expression across two experimental conditions, two genotypes, and neurons that project to two regions. I have a number of concerns about the experimental design and analysis. In sum, the data, as presented, do not fully support the claim that mPFC-projecting vHIP neurons are selectively activated during social encounters.

The rationale for using the behavioral task in Figure 1C is a bit unclear. (If this is a commonly used and well-accepted protocol, please cite its prior use.) On the control arm of the task, the subject gets two novel objects consecutively (two fake mice). On the experimental arm, the subject gets a familiar mouse and then a novel mouse. It is not clear why two steps are needed here. It seems one of the following designs may better assess social encounters:

a) Two arms being a familiar mouse, then sac, or a novel mouse, then sac.b) Two arms being a novel fake mouse, then sac, or a novel mouse, then sac.

The methods for quantifying images (as in Figure 1D) are not described.

The authors report number of cells as "n" in Figure 1E, but in this case it seems as though the correct biological replicate here should be number of animals (and c-Fos expression should be averaged within animal).

Statistically, the authors report a Kruskal-Wallis test, which is a non-parametric one-way ANOVA. But a one-way ANOVA is not appropriate for such an ambitious experiment, where three conditions are being compared simultaneously (genotype, behavioral condition, and projection). For example, the conclusion that "the differences in c-Fos intensities between the object and social conditions were smaller in Mecp2 KO mice" (subsection “mPFC-projecting vHIP neurons are selectively activated during social encounters”) appears to be true, but is not supported by a statistically significant interaction between genotype and behavioral condition. Overall, the study is likely under-powered (2-4 animals per group) for a 3x2 design and a three-way ANOVA.

I have a similar concern for Figure 2. On the whole, the data in Figure 2 seem to suggest impairments in social behavior and social memory in Rett mice, but it is not clear if these findings are supported statistically. The authors conclude that "Mecp2 KO mice spent similar amounts of time in both chambers" (Subsection “Atypical social behavior and impaired social memory in *Mecp2* KO mice”), whereas WT mice spent more time in the chamber with stranger 2." This result, compared with the left panel of Figure 2C, leads the authors to conclude that Rett mice have normal social preference, but impaired social memory. However, paired t-tests are not sufficient to make across-group comparisons. A two-way RM ANOVA would be needed for both the left and right panels in Figure 2C. A statistically significant interaction in the right panel would provide statistical support to the claim that Rett mice have impaired social memory. The same point also applies to Figure 2G.

Subsection “Atypical social behavior and impaired social memory in *Mecp2* KO mice”: "During the first four minutes of the assay, WT mice preferentially interacted with the novel mouse across all measured social behaviors, including following, sniffing, and "jumping on", after which social interaction declined regardless of target (n=6 mice, p<0.05, Student's paired t-test Figure 2F)." Two questions here:

a) It is not clear what the statistics are referring to here. It could be either the finding that WT mice interacted with the novel mouse preferentially in social behaviors (in which case the callout should be 2G), or it could be the claim that social interaction declined after the first four minutes. If the latter, the summarized data should be plotted in a panel in addition to the example trace in Figure 2F.

b) Why do the authors extract the first 4 minutes out of 10 for analysis? Has this been done by previous groups, and/or was this cutoff determined a priori? This is an important cutoff to justify, as it is used in multiple analyses throughout the paper.

Overall, Figure 3 nicely demonstrates that vHIP stimulation drives increased excitability in mPFC slices, but intracortical stimulation suggests that the local network is hypoexcitable in Rett. I have only a few minor questions about Figure 3:

The authors claim that VSD signals in ex vivo slices are directly proportional to the amplitude of field EPSPs and follow their kinetics (subsection “Increased influence of vHIP input on the mPFC network in *Mecp2* KO mice”, Figure 3B-C). The only demonstration of this is a single example traces in Figure 3C. This point would be made much more strongly with an example I/O curve.

How is dF/F calculated for VSD signals in I/O curves (such as Figure 3E)? I apologize if I missed this in the methods. The example in Figure 3D includes both a time dimension and a spatial dimension of the VSD signal. Do the points in Figure 3E represent summated dF/F in the whole window at the peak time? Or the most intense peak in the window at the peak time?

In Figure 3—figure supplement 2, the authors demonstrate that Rett slices have impaired LTP. They present normalized data, likely because the Rett slices are hyperactive at baseline (visible in example trace and demonstrated in Figure 3). Is it possible that the lack of LTP might reflect a ceiling effect?

Figures 4—figure supplement 1, Figure 4—figure supplement 2, Figure 4—figure supplement 3 are mentioned in the text as being related to Figure 4, but they seem more related to Figure 3. These data show that social* and electrophysiological** impairments in Rett mice are not seen at the presymptomatic stage. This is an important point, and could perhaps even be highlighted in its own section in the text.

Figure 4—figure supplement 1Figure should use RM two-way ANOVAs. Also related to Figure 4—figure supplement 1Figure: do presymptomatic mice also show impaired social behavior in the unrestricted social assay?

Figure 4—figure supplement 3Figure should use two-way ANOVAs to compare genotype and age. A significant interaction between genotype and age would be needed to make the claim in subsection “Selective chemogenetic manipulation of mPFC-projecting vHIP neurons regulates social memory”.

Figure 4C-D presents another ambitious study: bidirectional DREADD manipulation of the vHIP -> mPFC pathway in WT and Rett mice. I have a few questions:

Were the experimenters blind to genotype for these experiments? If so, why is there an excitatory DREADD group only for the WT condition?

Student's and nonparametric paired t-tests are not appropriate to compare three conditions: genotype, familiar vs. novel, and DREADD. Therefore it is difficult to assess whether inhibitory DREADDs do indeed improve performance in Rett mice, or whether excitatory DREADDs do indeed mimic altered performance. One idea that perhaps could reduce the number of dimensions from 3 to 2 for this and other studies would be to plot "time following" as a proportion within animal (similar to "memory discrepancy" in Figure 4I). This way, a 2-way ANOVA would work instead of a 3-way ANOVA. Then perhaps a supplement could show the absolute following times in a descriptive way to the familiar and novel animal, because it is nice to see those raw data as well.

The way the electrophysiological data are presented in Figure 4 appear much cleaner, but again, these experiments were done at the same time. It is not clear how statistics were done here: were two-way ANOVAs done individually on each panel in Figures 4F and 4H? Because each panel contains WT mCherry and KO mCherry, plus a different condition. I like that it is plotted this way to make things easier to understand, but it is not clear whether the statistical analysis is done on the entire dataset (genotype X DREADD X stim. intensity)? Again, perhaps the stim. intensity dimension could be collapsed if need be to a single value if a three-way test is prohibitive.

Figure 4I is potentially very interesting, showing a within-animal correlation between behavioral performance and long-range electrophysiology, three days later. But this is only shown for the Mecp2 KO + inhibitory DREADD group. Is this correlation seen in other experimental groups?

In Figure 4K, what is the cutoff to decide if an animal has "improved social memory" or "no memory"?

The statistical questions presented earlier also apply to Figure 5 and make it difficult to interpret these results.

In Figure 6, the "n" used is not clear – the authors report an "n" of 9 sections from 3 mice. But the figures show only 3 data points. It seems as though mice would be more appropriate here than sections, and sections can be pooled.

Similar statistical concerns about the lack of two-way ANOVA for Figure 7, though it is clear there is no difference in Figure 7F. Statistical comparisons in 7G-H also should account for multiple comparisons.

Figure 8 nicely demonstrates what would be expected based on the results in Figure 3. The electrophysiological results overall are very convincing between these two figures: hyperactive long-range vHIP->mPFC and hypoactive local or c-mPFC->mPFC. Figure 8 also extends this result to show that long-range hyperexcitability is seen in excitatory neurons but not inhibitory neurons. However, the sample size is much larger in pyramidal cells (n=15 or n=11, versus n=6 in inhibitory cells in 8D). Increasing the sample size to ~10-15 neurons for Figure 8D-H would be useful in helping to understand if the trend seen in 8D and 8H (p=0.06) would reach statistical significance with equivalent power.

*Reviewer #3:*

This is an excellent, and quite thorough, body of work analyzing the role of the ventral hippocampus to medial prefrontal cortex circuit in social behavior, and specifically looking at the role in mice with MECP2 mutations. I think the work stands nicely on its own and provides a clear story. I have no major concerns and do not feel any additional experiments are needed.

A minor concern is that the statistical testing performed should be consistent. In some of the behavior analysis, non-parametric tests (Wilcoxon, Mann-Whitney, etc) are used and in other cases parametric (t-test, anova) are used. This is problematic when there are different tests used for different components of the same overall analysis, for example in Figure 2E for free moving behavior all the comparisons used student t-test except for aggressor which used Kolomogorov-Smirnov testing, which is probably not an appropriate test in this case anyway. It would appear that the test was cherry picked to identify significance, rather than choosing appropriate statistical methods before analysis. A non-parametric rank test like Mann-Whitney is probably appropriate, and ultimately if no significance exists for aggressor it will in no way change the overall manuscript.

A question is why behavior scores for social memory assessment were only taken for the first 4 minutes. Some rationale, hopefully generated a priori, for restricting the analysis to the first 4 minutes should be provided.

---

## [Author Response]

Essential revisions:1) As stated in the individual reviewers' comments below, statistical analysis in each figure should strengthened. In particular the authors should present better justifications for the statistical tests they used and also validate the sample sizes (i.e., perform power analyses). If all claims hold up under statistical scrutiny, then many of the other criticisms in the individual reviews become minor.

We appreciate and reciprocate the reviewers’ goals to ensure that the proper statistical tests were used for the comparisons of all data sets. To this extent, we consulted with the Biostatistics Clinic of UAB’s Center for Clinical and Translational Science, and followed their advice to address the reviewers’ questions, and to re-analyze all data sets in the manuscript. As stated below for each case, these analyses did not change the conclusions reached in the original manuscript.

2) The authors should comment on the potential caveats of their experimental design in several places (use of cFOS signals and long term use of DREADD-CNO system). Experimental design details require better justification overall.

We have addressed the potential caveats of the long-term expression of DREADD and CNO administration, and better described the rationale for using c-Fos as a surrogate of neuronal activity. In addition, we now include additional details of experimental design in the Materials and methods section.

3) Related to the above point, the design of the social interaction tests requires justification. It is unclear why they authors chose to only assess the first 4 minutes of the 10 minute encounter for the behavioral test. A reasonable rationale should be provided to explain this choice.

In the revised manuscript and below, we provide an explanation for the use of the first 4 minutes of social interactions. In addition, we include a new supplementary figure to justify this analysis design.

The individual reviews are included below for your information.

Reviewer #1:

This study uses a large array of techniques to examine the properties of vHIP – mPFC connections in wild type and MeCP2KO animal to establish the role of this circuit in regulation of social behavior. Although a large number of synaptic and cellular deficits have been characterized in RTT models, the circuit deficits associated with this disorder remain poorly understood. Therefore, the current study is in principle a welcome addition to existing canon of studies on RTT pathophysiology. However, the authors need to clarify several aspects of their experiments and provide more technical details to make this rather extensive study more accessible to a wider audience.1) Figure 1: cFOS staining should use histograms to better present the distribution of fluorescence intensities? What do these intensities present, cells, clusters? Clarify the relationship between neuronal activity and cFOS staining. Why count neurons, rather than quantify staining intensity per neuron?

We apologize for our lack of clarity: as reviewer 1 suggests, we have quantified the cFos intensity per neuron. In the original submission, each “n” in the bar graphs was an individual neuron. The new panel D in Figure 1 shows the cumulative probability distribution of c-Fos intensity per neuron, which presents the same information as a histogram but without relying on the statistical mean. In the new panel E of Figure 1 each “n” is an individual mouse, which allowed performing a 3-Way ANOVA, as requested by reviewer 2. We now extended the description of this experiment in the Results section and Materials and methods section of the manuscript.

Our rationale to quantify c-Fos intensity per neuron instead of counting the number of c-Fos-positive neurons is based on the fact that *c-fos* expression is a graded reflection of neuronal activity mediated by different intracellular Ca^2+^ levels (Cohen and Greenberg, 2009). In addition, we consider that choosing an arbitrary c-Fos intensity value as a threshold cut-off is more biased than presenting all c-Fos intensity values displayed by all neurons. Note that every c-Fos-positive cell was confirmed to be a neuron based on its NeuN staining, and that their projection target identified by retread labeling (i.e. three fluorescence channels: dual immunostaining for c-Fos and NeuN, plus retread fluorescence).

2) Figure 3: The relationship between fEPSPs and voltage sensitive dye signals. EPSP slope is a more reliable reporter for synaptic responses, whereas fEPSP amplitudes are rather non-linear reporters for postsynaptic activity. What do the authors think they are monitoring using voltage sensitive dyes (synaptic responses, firing or a loosely defined measure of "activity")?

The VSD signals reported in our manuscript occur during single subthreshold excitatory postsynaptic potentials recorded with extracellular electrodes (field EPSPs), not action potential firing, which is evident as faster rising and larger population spikes. We use this VSD imaging approach to estimate the depolarization evoked throughout the network by stimulation of selected afferent fibers. VSD imaging is complementary to electrophysiological recordings because it provides spatial information from sites beyond the recording site (e.g. Chang and Jackson, 2003).

We agree with the reviewer that the slope of the field EPSP reflects more accurately the underlying postsynaptic current without confounding dendritic depolarizations. In addition to following their kinetics and being directly proportional to the amplitude of field EPSPs in input-output curves, the ∆F/F of evoked VSD signals is also directly proportional to the initial slope of field EPSPs.The peak of VSD aptitudes reflects the largest synaptic depolarization throughout the network. The spatial spread of VSD signals reflects the proportion of the network with postsynaptic responses at a set afferent stimulus strength, while the temporal spread combines the proportion of the responding network with the duration of postsynaptic depolarization across the cortical network. We now include the figure below as a supplementary figure to better describe the relationship between VSD signals and postsynaptic responses, and endeavor to make these concepts more clear in the revised manuscript

3) The authors should be commended for the well-controlled DREADD-CNO experiments. That said, the authors primarily perform long term experiments using this system. It is important to validate the DREADD system works as expected under the conditions of long-term (11 days?) exposure to CNO. It is plausible to expect some level of adaptation and plasticity in the system. However, the authors' interpretation of their result does not take this possibility into account.

We apologize for our lack of clarity describing our rationale for these experiments. Neurons expressing the excitatory DREADD hM3Dq in slices from mice exposed to CNO for 14 days still show an increased firing rate after CNO application (Cheng et al., 2018). We now include this rationale and its justification in the revised manuscript.

4) It is extremely difficult to interpret the experiments presented in Figure 8 in terms of a change in presynaptic efficacy. The authors use an extremely artificial setting (4-AP + TTX+ 4 mM Ca^2+^) to evaluate synaptic efficacy via optogenetic stimulation. It is difficult ascertain if the differences seen in these experiments are indeed due to bigger boutons and increased presynaptic release probability.

Regarding the issue of synaptic strength, we did not intend to interpret our results as reflecting a purely presynaptic difference between *Mecp2* KO and wild-type mice.The larger volume of presynaptic boutons of vHIP axons in the mPFC of *Mecp2* KO mice suggest higher synaptic strength, but not necessarily due solely to a presynaptic mechanism. Larger presynaptic boutons are correlated to larger active zones and thus higher numbers of docked vesicles and higher Pr (e.g. Murthy et al., 2001), but also to larger spine heads with larger postsynaptic densities containing more glutamate receptors (Matsuzaki et al., 2004). The data presented in Figure 8 supports a difference in synaptic strength due to a postsynaptic mechanism:higher AMPA/NMDA ratio at vHIP-evoked optical EPSCs in mPFC layer 5 pyramidal neurons from *Mecp2* KO mice. The observation of larger presynaptic boutons and higher AMPA/NMDA ratios reflect mechanisms underlying synaptic strength that are not mutually exclusive. In addition, preliminary analyses of optical EPSCs evoked by paired-pulse stimulation in standard recording conditions (i.e. normal extracellular Ca^2+^, no TTX, no 4-AP) suggest a higher Pr at vHIP synapses on mPFC pyramidal neurons in *Mecp2* KO mice, but not on interneurons. The parsimonious interpretation is that vHIP synapses on mPFC pyramidal neurons are stronger in *Mecp2* KO mice due to both higher presynaptic efficacy and higher postsynaptic responsiveness. We consider that the full characterization of presynaptic and postsynaptic function at vHIP-mPFC synapses is beyond the scope of the current manuscript.

We apologize for our lack of clarity describing our rationale for the design of these experiments. The reason to perform optogenetic stimulation in the presence of TTX, 4-AP, and 4mM extracellular Ca^2+^ was to isolate monosynaptic EPSCs for accurate measurements of their amplitude without the confounding contribution of polysynaptic responses (Petreneau et al., 2009).

Reviewer #2:The authors demonstrate that the projection from ventral hippocampus to mPFC is hyperactive in symptomatic Rett mice, yet activity driven by intracortical stimulation is decreased. The electrophysiological findings in Figure 3 and Figure 4, and anatomical findings in Figure 6, are convincing and compelling. However, the link to behavior is tenuous, largely because of issues with the statistics. It is difficult to gauge the claims made in Figure 1, Figure 2, Figure 4, Figure 5. Although this is a potentially very interesting manuscript, the statistical issues make it difficult to assess the impact of the paper, as well as what additional experiments or analyses may be needed.Specifically:Figure 1 represents an extremely ambitious experiment, as the authors compare c-Fos expression across two experimental conditions, two genotypes, and neurons that project to two regions. I have a number of concerns about the experimental design and analysis. In sum, the data, as presented, do not fully support the claim that mPFC-projecting vHIP neurons are selectively activated during social encounters.The rationale for using the behavioral task in Figure 1C is a bit unclear. (If this is a commonly used and well-accepted protocol, please cite its prior use.) On the control arm of the task, the subject gets two novel objects consecutively (two fake mice). On the experimental arm, the subject gets a familiar mouse and then a novel mouse. It is not clear why two steps are needed here. It seems one of the following designs may better assess social encounters:a) Two arms being a familiar mouse, then sac, or a novel mouse, then sac.b) Two arms being a novel fake mouse, then sac, or a novel mouse, then sac.

We thank the reviewer for this opportunity to clarify the rationale of the these experiments. The two experimental designs suggested by the reviewer address two different issues: (a) is designed to test the hypothesis that vHIP neurons are activated by social interaction with a novel mouse (as opposed to a familiar one), while (b) tests the hypothesis that vHIP neurons are activated by social interaction with a novel mouse (as opposed to a toy mouse).

Our experimental design aims to test the second issue: whether mPFC-projecting vHIP neurons are activated by a social encounter, as opposed to exposure to a toy mouse. Okuyama et al., (2016) describe that exposure to mice caused activation of vHIP neurons, regardless of novelty; however, these authors did not differentiate between vHIP neurons that project to different brain areas. Because we were unsure of the sensitivity of the c-Fos intensity assay, we choose to present a strong stimulation to recruit as many mPFC-projecting vHIP neurons as possible to increase the signal-tonoise of the assay. To ensure maximal activation of as many mPFC-projecting vHIP neurons as possible we performed sequential interactions, first with a littermate mouse followed by a non-littermate mouse 1 hour later as the "social condition” Please note that this experiment was not performed to identify the vHIP neurons that encode social memory (as in the reviewer’s (a) design), which we think requires the application of a different experimental approach other than just c-Fos labeling.

The methods for quantifying images (as in Figure 1D) are not described.

We apologize for this oversight, we now include all methodological details.

The authors report number of cells as "n" in Figure 1E, but in this case it seems as though the correct biological replicate here should be number of animals (and c-Fos expression should be averaged within animal).Statistically, the authors report a Kruskal-Wallis test, which is a non-parametric one-way ANOVA. But a one-way ANOVA is not appropriate for such an ambitious experiment, where three conditions are being compared simultaneously (genotype, behavioral condition, and projection). For example, the conclusion that "the differences in c-Fos intensities between the object and social conditions were smaller in Mecp2 KO mice" (subsection “mPFC-projecting vHIP neurons are selectively activated during social encounters”) appears to be true, but is not supported by a statistically significant interaction between genotype and behavioral condition. Overall, the study is likely under-powered (2-4 animals per group) for a 3x2 design and a three-way ANOVA.

Following the suggestion from reviewer 1 and consultation with biostatisticians at UAB (see above), panel D in Figure 1 in the revised manuscript shows the cumulative probability distributions of c-Fos intensities from individual neurons. In addition, panel D in the new Figure 1 shows the same data as bar graphs of c-Fos intensity in each cell averaged per mouse as the sampling number “n”. We performed a three-way ANOVA using genotype, behavioral condition, and projection as the conditions, which yielded statistically significant differences between the groups as in the original analyses.

I have a similar concern for Figure 2. On the whole, the data in Figure 2 seem to suggest impairments in social behavior and social memory in Rett mice, but it is not clear if these findings are supported statistically. The authors conclude that "Mecp2 KO mice spent similar amounts of time in both chambers" (Subsection “Atypical social behavior and impaired social memory in Mecp2 KO mice”), whereas WT mice spent more time in the chamber with stranger 2." This result, compared with the left panel of Figure 2C, leads the authors to conclude that Rett mice have normal social preference, but impaired social memory. However, paired t-tests are not sufficient to make across-group comparisons. A two-way RM ANOVA would be needed for both the left and right panels in Figure 2C. A statistically significant interaction in the right panel would provide statistical support to the claim that Rett mice have impaired social memory. The same point also applies to Figure 2G.

Following reviewer 2’s suggestion in Figure 4 question 2, we have changed all the panels illustrating the social memory results to now show a memory Discrimination Index as the variable for statistical comparisons, which yields a single value per group. The memory Discrimination Index is the difference in the time spent following a novel mouse compared to the time spent following a familiar mouse. For comparisons between 2 groups, we used either a Student’s t-test for parametric data or a MannWhitney test for non-parametric data (data distribution was tested with the D’Agostino and Pearson’s test for normalcy). For comparisons between 3 or more groups, we used a two-way ANOVA. In addition, we now report a post-hoc one sample t-test comparing the Discrimination Index values against chance values to determine if each group has a significant preference in following times. In addition, we now include all the raw data of following times in supplemental figures.

Subsection “Atypical social behavior and impaired social memory in Mecp2 KO mice”: "During the first four minutes of the assay, WT mice preferentially interacted with the novel mouse across all measured social behaviors, including following, sniffing, and "jumping on", after which social interaction declined regardless of target (n=6 mice, p<0.05, Student's paired t-test Figure 2F)." Two questions here:

We apologize for the typographical mistake: the statistics reported in 2G are about the memory component.

The rationale of using the first 4 minutes of the total 10 minutes of the unrestricted social interaction assay is based on the observation that more than 75% of the social interactions occur during the first 4 minutes. After that time, there is no longer a difference in time spent interacting with the target the mice compared to the time spent exploring the environment, as the test mouse becomes acclimated to their presence. As a result of this, the Discrimination Index after the first 4 minutes become a less reliable estimate of the social interactions. The entirety of the 10 minute videos are used for quantification of non-target specific behaviors, as different behaviors emerged at later times in the movies. We have plotted the analyses of the entire 10-minute videos and included as Figure 2—figure supplement 2.

a) It is not clear what the statistics are referring to here. It could be either the finding that WT mice interacted with the novel mouse preferentially in social behaviors (in which case the callout should be 2G), or it could be the claim that social interaction declined after the first four minutes. If the latter, the summarized data should be plotted in a panel in addition to the example trace in Figure 2F.b) Why do the authors extract the first 4 minutes out of 10 for analysis? Has this been done by previous groups, and/or was this cutoff determined a priori? This is an important cutoff to justify, as it is used in multiple analyses throughout the paper.Overall, Figure 3 nicely demonstrates that vHIP stimulation drives increased excitability in mPFC slices, but intracortical stimulation suggests that the local network is hypoexcitable in Rett. I have only a few minor questions about Figure 3:

*The authors claim that VSD signals in* ex vivo *slices are directly proportional to the amplitude of field EPSPs and follow their kinetics (subsection “Increased influence of vHIP input on the mPFC network in Mecp2 KO mice”, Figure 3B-C). The only demonstration of this is a single example traces in Figure 3C. This point would be made much more strongly with an example I/O curve.*

Please see our answer to question 2 by reviewer 1. Briefly, we now include the input output relationships of VSD signals and the initial slope of field EPSPs in a

supplementary figure, including representative images of color coded ∆F/F ratios, and ∆F/F traces from 4 stimulation intensities. Similar direct proportionality has been consistently reported in papers using organic VSD dyes for imaging subthresold voltage responses in ex vivo brain slices (e.g. Chang and Jackson, 2003; Bandyopadhyay et al., 2005; Calfa et al., 2011).

How is dF/F calculated for VSD signals in I/O curves (such as Figure 3E)? I apologize if I missed this in the methods. The example in Figure 3D includes both a time dimension and a spatial dimension of the VSD signal. Do the points in Figure 3E represent summated dF/F in the whole window at the peak time? Or the most intense peak in the window at the peak time?

We apologize for the lack of detail in the Materials and methods section, which are now included in the revised manuscript. The maximum VSD signals are the mean ∆F/F ratios within a 3x3 pixel region of interest (ROI) placed on the location with the highest ∆F/F ratio at 6 ms after afferent stimulation, which always occurred within layer 2/3 of mPFC slices. We chose 6 ms after stimulation because that coincides with the peak of the extracellularly recorded field EPSP. We now have uploaded our code to the GitHub repository and provide the link in the manuscript.

In Figure 3—figure supplement 2, the authors demonstrate that Rett slices have impaired LTP. They present normalized data, likely because the Rett slices are hyperactive at baseline (visible in example trace and demonstrated in Figure 3). Is it possible that the lack of LTP might reflect a ceiling effect?

The reviewer is correct, we think that the lack of LTP at vHIP-mPFC synapses is due to a “ceiling” effect. We have demonstrated that hippocampal CA3-CA1 synapses of *Mecp2* KO mice are potentiated at baseline due to impaired synaptic AMPAR trafficking, which prevents further LTP (Li et al., 2016); a similar mechanism impairs homeostatic synaptic plasticity in *Mecp2* KO hippocampal neurons in culture (Xu and Pozzo-Miller, 2017). We consider that performing the long list of experiments required to demonstrate the same mechanism at vHIP-mPFC synapses is beyond the scope of the present manuscript, and is the focus of our immediate efforts in the lab.

Figure 4—figure supplement 1, Figure 4—figure supplement 2, Figure 4—figure supplement 3 are mentioned in the text as being related to Figure 4, but they seem more related to Figure 3. These data show that social* and electrophysiological** impairments in Rett mice are not seen at the presymptomatic stage. This is an important point, and could perhaps even be highlighted in its own section in the text.

We thank the reviewer for this suggestion. We agree that this is an important observation because it confirms other reports of typical behaviors and synaptic function in young presymptomatic *Mecp2* KO mice (including our own work in the hippocampus), and have described it in the Results section, including a supplemental figure. However, presenting these data in a main figure will significantly lengthen the manuscript and distract from the main conclusions we wish to present.

Figure 4—figure supplement 1 should use RM two-way ANOVAs. Also related to Figure 4—figure supplement 1: do presymptomatic mice also show impaired social behavior in the unrestricted social assay?

We thank the reviewer for this suggestion. We consider these experiments beyond the scope of this manuscript, because they are better suited for a separate study of the postnatal development of the vHIP-mPFC projection and its role in social behaviors, as well as of the developmental progression of social behavior deficits in *Mecp2* KO mice.

Figure 4—figure supplement 3should use two-way ANOVAs to compare genotype and age. A significant interaction between genotype and age would be needed to make the claim in subsection “Selective chemogenetic manipulation of mPFC-projecting vHIP neurons regulates social memory”.

We thank the reviewer for this suggestion. We have reanalyzed these data using a twoway ANOVA and find a significant interaction between Age and Genotype, with the outcome of the multiple comparisons analyses matching our original results.

Figure 4C-D presents another ambitious study: bidirectional DREADD manipulation of the vHIP -> mPFC pathway in WT and Rett mice. I have a few questions:Were the experimenters blind to genotype for these experiments? If so, why is there an excitatory DREADD group only for the WT condition?

We decided to only test the effects of the excitatory DREADD in WT mice because making the vHIP more hyperactive in *Mecp2* KO mice will surely affect behavioral responses that are already affected. Since the outcome of these experiments may not reveal novel information about Rett syndrome, we decided not to use more animals than those strictly necessary. The experimenters were not blinded to genotype when injecting mice in order to optimize the number of animals to be used in these experiments. The experimenters were blinded during behavioral testing, though some *Mecp2* KO mice were profoundly symptomatic and easy to identify. The implementation of computer vision to score all behavioral data, and of Matlab codes to analyze all VSD data are additional steps taken to ensure unbiased acquisition and analyses of data.

Student's and nonparametric paired t-tests are not appropriate to compare three conditions: genotype, familiar vs. novel, and DREADD. Therefore it is difficult to assess whether inhibitory DREADDs do indeed improve performance in Rett mice, or whether excitatory DREADDs do indeed mimic altered performance. One idea that perhaps could reduce the number of dimensions from 3 to 2 for this and other studies would be to plot "time following" as a proportion within animal (similar to "memory discrepancy" in Figure 4I). This way, a 2-way ANOVA would work instead of a 3-way ANOVA. Then perhaps a supplement could show the absolute following times in a descriptive way to the familiar and novel animal, because it is nice to see those raw data as well.The way the electrophysiological data are presented in Figure 4 appear much cleaner, but again, these experiments were done at the same time.

Regarding the changes made to the statistical analyses of behavioral data, please refer to our response to Figure 2 question 1. To briefly restate, following advice from biostatisticians at UAB (see above) and the reviewer’s suggestion, we have consolidated the time spent following familiar and novel mice into a Discrimination Index; the raw data of following times is now presented in supplementary figures. We performed a two-way ANOVA (Group x Discrimination Index) with multiple comparisons and a post-hoc one-sample t-test against chance to determine significant preferences. The outcome of these tests match the results presented in the original manuscript. We could not perform a three-way ANOVA (Genotype x DREADD x Memory) because the 2 genotypes did not receive the same DREADD treatment (as justified above). We have therefore referred to the Genotype x DREADD as “Group”, as per the suggestion of biostatisticians at UAB.

It is not clear how statistics were done here: were two-way ANOVAs done individually on each panel in Figures 4F and 4H? Because each panel contains WT mCherry and KO mCherry, plus a different condition. I like that it is plotted this way to make things easier to understand, but it is not clear whether the statistical analysis is done on the entire dataset (genotype X DREADD X stim. intensity)? Again, perhaps the stim. intensity dimension could be collapsed if need be to a single value if a three-way test is prohibitive.

We apologize for our lack of clarity. We performed a two-way repeated measures ANOVA (Group X Stimulation Intensity, with Stimulation Intensity being the repeated measure). We cannot do a three-way ANOVA because the genotypes received different DREADD treatments as stated above.

Figure 4I is potentially very interesting, showing a within-animal correlation between behavioral performance and long-range electrophysiology, three days later. But this is only shown for the Mecp2 KO + inhibitory DREADD group. Is this correlation seen in other experimental groups?

We now include the analyses of within-animal correlation of social memory

Discrimination Index with the amplitude of vHIP-evoked VSD signals in mPFC slices in the other experimental groups. Unfortunately, we have only 3 surviving control *Mecp2* KO mice expressing mCherry that yielded both social behaviors and VSD imaging data in ex vivo slices. However, this group shows a trend of a correlation of vHIP-evoked VSD signals and memory Discrimination Index similar to that of *Mecp2* KO mice expressing the inhibitory DREADD. When these data from all *Mecp2* KO mice are pooled, there is a statistically significant correlation between social memory and vHIPdriven mPFC activity. Interestingly, this correlation is not statistically significant in any of the WT groups, which suggest an intriguing contribution of the altered mPFC microcircuit in *Mecp2* KO mice. The correlation between social memory and mPFC activity evoked by intra-cortical stimulation is not statistically significant in any of the groups. The figure is included in the revised manuscript as Figure 4—figure supplement 5.

In Figure 4K, what is the cutoff to decide if an animal has "improved social memory" or "no memory"?

Throughout the manuscript we used a social memory Discrimination Index (DI), which was calculated as:

DI = (t_following unknown_ – t_following known_/ t_following unknown_ + t_following known_) * 100

Using this ratio, “Improved social memory” is DI > 0, and “No memory” is DI ≤ 0. This information has been added to the Materials and methods section.

The statistical questions presented earlier also apply to Figure 5 and make it difficult to interpret these results.

Please see our responses above regarding to new statistical analyses of behavioral data. Comparisons of the Discrimination Index between groups yielded the same outcome for the acute *Mecp2* group and the NAc projection group. However, the DI was not statistically different between controls and hM3Dq-expressing WT mice in the mPFC projections group due to the large variability in the hM3Dq group. The control group had a DI significantly different than chance, whereas the hM3Dq did not. These new results do not change the main conclusion of our study.

In Figure 6, the "n" used is not clear – the authors report an "n" of 9 sections from 3 mice. But the figures show only 3 data points. It seems as though mice would be more appropriate here than sections, and sections can be pooled.

We apologize for our lack of clarity. The reviewer is correct, the brain sections were pooled and individual data points are the statistical averages across mice.

Similar statistical concerns about the lack of two-way ANOVA for Figure 7, though it is clear there is no difference in Figure 7F. Statistical comparisons in 7G-H also should account for multiple comparisons.

We moved the data of vHIP presynaptic boutons within mPFC Layer 2/3 to a supplemental figure. In addition, we performed a two-way ANOVA (Genotype x Projection) for the comparison of vHIP presynaptic boutons. We now show both the cumulative probability of vHIP bouton sizes and a bar graph with the statistical averages. We performed a two-way ANOVA to compare these data, which yielded the same outcome as in the original results.

Figure 8 nicely demonstrates what would be expected based on the results in Figure 3. The electrophysiological results overall are very convincing between these two figures: hyperactive long-range vHIP->mPFC and hypoactive local or c-mPFC->mPFC. Figure 8 also extends this result to show that long-range hyperexcitability is seen in excitatory neurons but not inhibitory neurons. However, the sample size is much larger in pyramidal cells (n=15 or n=11, versus n=6 in inhibitory cells in 8D). Increasing the sample size to ~10-15 neurons for Figure 8D-H would be useful in helping to understand if the trend seen in 8D and 8H (p=0.06) would reach statistical significance with equivalent power.

Following the reviewer’s suggestion, we have increased the sample size of the wholecell recordings from mPFC interneurons during optical stimulation of either vHIP or contralateral mPFC axons to n=11 from WT mice and n=12 from *Mecp2* KO mice. Statistical comparisons yielded no significant differences in either vHIP-evoked synaptic currents or mPFC-evoked synaptic currents, as well as in the normalized data. We also have a larger sample size of AMPA/NMDA ratio evoked by optical stimulation of vHIP axons in mPFC interneurons, which also show no significant differences between WT and Mecp2 KO mice. These results with larger sample sizes yielded the same outcome as in the original results.

Reviewer #3:This is an excellent, and quite thorough, body of work analyzing the role of the ventral hippocampus to medial prefrontal cortex circuit in social behavior, and specifically looking at the role in mice with MECP2 mutations. I think the work stands nicely on its own and provides a clear story. I have no major concerns and do not feel any additional experiments are needed.A minor concern is that the statistical testing performed should be consistent. In some of the behavior analysis, non-parametric tests (Wilcoxon, Mann-Whitney, etc) are used and in other cases parametric (t-test, anova) are used. This is problematic when there are different tests used for different components of the same overall analysis, for example in Figure 2E for free moving behavior all the comparisons used student t-test except for aggressor which used Kolomogorov-Smirnov testing, which is probably not an appropriate test in this case anyway. It would appear that the test was cherry picked to identify significance, rather than choosing appropriate statistical methods before analysis. A non-parametric rank test like Mann-Whitney is probably appropriate, and ultimately if no significance exists for aggressor it will in no way change the overall manuscript.

We apologize for our lack of clarity and for giving the impression that we selected statistical tests to hunt for p value significance (not our practice). All statistical tests in the revised manuscript have been deemed appropriate by experts of the Biostatistics Clinic of UAB’s Center for Clinical and Translational Science. For all statistical comparisons, we first tested if the data fitted a normal distribution using either the D’Agostino and Pearson’s test or the Shapiro-Wilk test when the number of replicates was below the requirement for D’Agostino & Pearson test. Depending on the result of this normalcy test, the data were compared using either a parametric (normal distribution) or a non-parametric test (non-normal distribution). We list all the statistical tests and the pvalues yielded in each comparison in the statistical table. We re-analyzed all nonparametric data sets with two samples using the Mann-Whitney test as suggested, which yielded a lack of statistical significance for the aggression phenotype shown in Figure 2E.

A question is why behavior scores for social memory assessment were only taken for the first 4 minutes. Some rationale, hopefully generated a priori, for restricting the analysis to the first 4 minutes should be provided.

The rationale of using the first 4 minutes of the total 10 minutes of the unrestricted social interaction assay is described above in our responses to reviewer 2, including a figure that is also a supplemental figure in the revised manuscript.